# Structural basis of promoter recognition by *Staphylococcus aureus* RNA polymerase

Linggang Yuan[1,5], Qingyang Liu[1,5], Liqiao Xu[1,5], Bing Wu[2,3] & Yu Feng [1,4] ✉

Bacterial RNAP needs to form holoenzyme with σ factors to initiate transcription. While *Staphylococcus aureus* σ$^A$ controls housekeeping functions, *S. aureus* σ$^B$ regulates virulence, biofilm formation, persistence, cell internalization, membrane transport, and antimicrobial resistance. Besides the sequence difference, the spacers between the −35 element and −10 element of σ$^B$ regulated promoters are shorter than those of σ$^A$ regulated promoters. Therefore, how σ$^B$ recognizes and initiates transcription from target promoters can not be inferred from that of the well studied σ. Here, we report the cryo-EM structures of *S. aureus* RNAP-promoter open complexes comprising σ$^A$ and σ$^B$, respectively. Structural analyses, in combination with biochemical experiments, reveal the structural basis for the promoter specificity of *S. aureus* transcription. Although the −10 element of σ$^A$ regulated promoters is recognized by domain σ$^A_2$ as single-stranded DNA, the −10 element of σ$^B$ regulated promoters is co-recognized by domains σ$^B_2$ and σ$^B_3$ as double-stranded DNA, accounting for the short spacers of σ$^B$ regulated promoters. *S. aureus* RNAP is a validated target of antibiotics, and our structures pave the way for rational drug design targeting *S. aureus* RNAP.

Bacterial RNA polymerase (RNAP) is the protein machinery responsible for transcription. Most bacterial RNAP is composed of five subunits-α$^I$, α$^{II}$, β, β′, and ω. The overall shape of bacterial RNAP resembles a crab claw, with the active center cleft located in the middle of two pincers[1]. During transcription initiation, the clamp, a mobile structural module that makes up much of one pincer, undergoes swing motions that open the active center cleft to allow entry of the promoter DNA[2–4]. During transcription elongation, the clamp closes up and secures the transcription bubble inside the active center cleft.

Bacterial RNAP forms holoenzyme with σ factors to initiate transcription[5]. Housekeeping σ factors (σ$^{70}$ in *E. coli* and σ$^A$ in other bacteria) govern the transcription of the majority of cellular genes. Housekeeping σ factors are comprised of several conserved domains: σ$_{1.1}$, σ$_{1.2}$, σ$_2$, σ$_3$, σ$_{3.2}$, and σ$_4$. For housekeeping σ factors, the consensus sequences of the promoter −35 element and −10 element are TTGACA and TATAAT, with an optimal spacer of 17 base pairs (bp). Extensive

genetic, biochemical and structural studies demonstrate that σ$^{70/A}_4$ contacts the flap tip helix (FTH) of the RNAP β subunit and recognizes the promoter −35 element as double-stranded DNA (dsDNA), while σ$^{70/A}_2$ contacts the clamp helices of the RNAP β′ subunit and recognizes the promoter −10 element as single-stranded DNA (ssDNA)[6–10]. In contrast to the housekeeping σ factors, alternative σ factors direct RNAP to specialized operons in response to environmental and physiological cues. For example, *Mycobacterium tuberculosis* σ$^H$ is a key regulator of the response to oxidative, nitrosative, and heat stresses[11]. For σ$^H$ regulated promoters, the consensus sequences of the −35 element and −10 element are GGAACA and GTT, with an optimal spacer of 17 bp. Similar to the housekeeping σ factors, the −35 element and −10 element are recognized by σ$^H_4$ and σ$^H_2$ as dsDNA and ssDNA, respectively[12]. σ$^{54}$, which is involved in a range of different stress responses, has no sequence similarity to housekeeping σ factors at all[13–17]. In contrast to σ$^{70}$ and σ$^H$, σ$^{54}$ is unable to unwind promoter DNA

[1]Department of Biophysics, and Department of Infectious Disease of Sir Run Run Shaw Hospital, Zhejiang University School of Medicine, Hangzhou, China. [2]Department of Gastroenterology and Hepatology, Minhang Hospital, Fudan University, Shanghai, China. [3]Institute of Fudan-Minhang Academic Health System, Minhang Hospital, Fudan University, Shanghai, China. [4]Key Laboratory for Diagnosis and Treatment of Physic-Chemical and Aging Injury Diseases of Zhejiang Province, Hangzhou, China. [5]These authors contributed equally: Linggang Yuan, Qingyang Liu, Liqiao Xu. ✉e-mail: yufengjay@zju.edu.cn

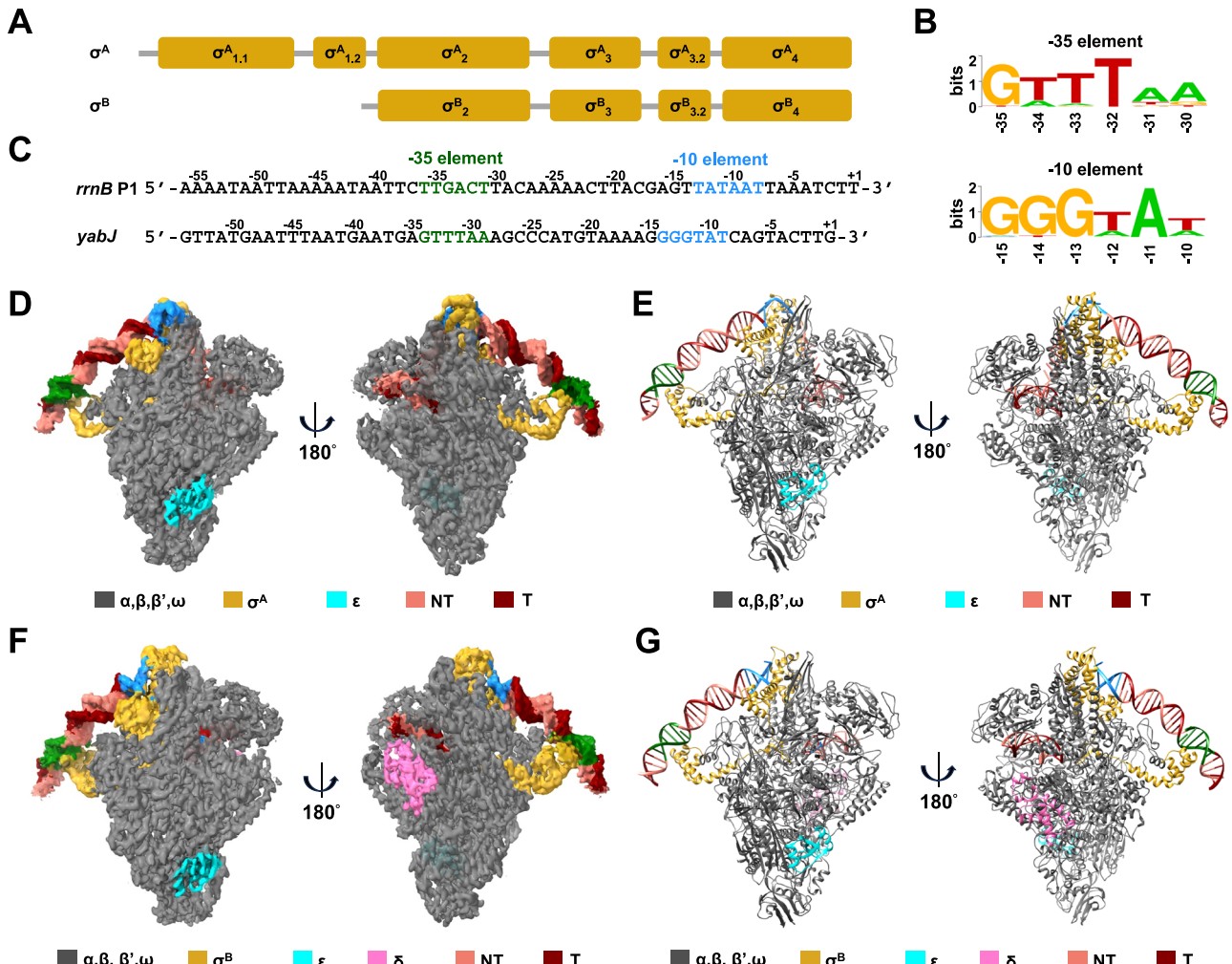

**Fig. 1 | The cryo-EM structures of σ^A^-RPo and σ^B^-RPo. A** Domain organization of *S. aureus* σ^A^ and σ^B^. **B** The consensus sequences of *S. aureus* σ^B^-dependent promoters. The sequence logo was created on the WebLogo website (http://weblogo.berkeley.edu) using an alignment of all the σ^B^-dependent promoters listed in Supplementary Table 1. The height of the letters is proportional to their frequency. **C** Sequences of *S. aureus rrnB* P1 and *yabJ* promoters, which are recognized by σ^A^ and σ^B^, respectively. Green, the −35 element; blue, the −10 element. **D, E** The electron potential map without B-factor sharpening (**D**) and the model (**E**) of σ^A^-RPo. Gray, RNAP core except ε; cyan, ε; yellow, σ^A^; salmon, nontemplate strand DNA; red, template strand DNA; green, the −35 element; blue, the −10 element. **F, G** The electron potential map without B-factor sharpening (**F**) and the model (**G**) of σ^B^-RPo. Gray, RNAP core except ε and δ; cyan, ε; pink, δ; yellow, σ^B^; salmon, nontemplate strand DNA; red, template strand DNA; green, the −35 element; blue, the −10 element.

spontaneously. Instead, it requires ATP dependent activator proteins bound upstream of the promoter in order to initiate transcription. The consensus sequences of the promoter −24 element and −12 element are TGGCACG and TTGCW (W = A/T), with an optimal spacer of 4 bp. σ^54^ recognizes the promoter −24 element and −12 element using RpoN and ELH-HTH domains, respectively.

σ^B^ was first discovered in *Bacillus subtilis*[18]. The activity of σ^B^ is tightly regulated by the Rsb proteins[19,20]. When there is no stress, RsbW binds to and sequesters σ^B^. Under stress conditions, RsbV binds to RsbW and releases σ^B^. Additionally, RsbU regulates the activity of σ^B^ by dephosphorylating RsbV. In *S. aureus*, σ^B^ is one of the major determinants of pathogenicity and virulence[21–23]. Although σ^A^ and σ^B^ share σ_2, σ_3, σ_{3.2}, and σ_4, the promoters of σ^B^ regulated genes show distinct signatures from those of σ^A^ regulated genes, ensuring the specificity of transcription regulation (Fig. 1A, B). First, the consensus sequence of the −35 element (GTTTWW) and −10 element (GGGWAW) are dramatically different from those of σ^A^ dependent promoters[21]. More importantly, the spacers between the −35 element and −10 element are divergent (~17 bp for σ^A^ *vs* ~14 bp for σ^B^).

Despite four decades of study, we still do not know how σ^B^ recognizes its promoters specifically and turns on transcription

efficiently. In this work, we solved the cryo-EM structures of *S. aureus* RNAP-promoter open complex comprising σ^A^ and σ^B^ (σ^A^-RPo and σ^B^-RPo), respectively. The structures define the interactions between RNAP holoenzyme and DNA, thus explaining the promoter specificity as well as the stabilization of transcription bubble.

## Results

To obtain *S. aureus* RNAP for structural study, we cloned genes encoding *S. aureus* RNAP α, β, β', ω, δ, and ε subunits into the pET21a vector and expressed the recombinant RNAP core enzyme in *E. coli* (Supplementary Fig. 1A, B). The activity of the RNAP core enzyme was verified using a primer extension assay on an RNA-DNA scaffold (Supplementary Fig. 1C). Then the RNAP holoenzyme was prepared by mixing the RNAP core enzyme with an excess of a σ factor and purified by size exclusion chromatography. In vitro transcription experiments confirmed that σ^A^-RNAP holoenzyme is active in transcribing from the *S. aureus rrnB* P1 promoter, a typical σ^A^ dependent promoter, and that σ^B^-RNAP holoenzyme is efficient in transcribing from the *S. aureus yabJ* promoter[21], a verified σ^B^ dependent promoter (Supplementary Fig. 1D).

To obtain the structure of σ^A^-RPo, we used a DNA scaffold modified from *S. aureus rrnB* P1 promoter, which contains a consensus −35

element and a consensus −10 element (Fig. 1C). To obtain the structure of σB-RPo, we used a DNA scaffold modified from *S. aureus yabJ* promoter. The scaffold contains a consensus −35 element and a consensus −10 element, as well. The formation of RPo was confirmed by electrophoretic mobility shift assay (EMSA, Supplementary Fig. 2). The structures of σA-RPo and σB-RPo were determined at 3.7 Å and 3.3 Å by cryo-EM single particle reconstruction, respectively (Supplementary Figs. 3–6 and Supplementary Table 2). The cryo-EM maps show unambiguous densities for α, β, β′, ω, δ, ε, σ, and the DNA scaffolds (Fig. 1 and Supplementary Fig. 7). The overall structure of *S. aureus* RNAP resembles those from other species with an overall shape of crab craw[1,24–29]. The clamp adopts a closed conformation, securing the transcription bubble and downstream dsDNA in the main channel (Supplementary Fig. 8).

There is only one insertion in *S. aureus* RNAP (βIn5, L281-K373, Supplementary Fig. 9). βIn5 inserts into and packs against the β lobe, resulting in an interface area of 1617 Å². The large interface area makes βIn5 and β lobe look like one whole domain. Since β lobe is the target of transcription factors[30–32], βIn5 may serve as the docking site for transcription factors. In *E. coli* RNAP, βSI1 inserts into the β lobe at a different site. The interface area of βSI1 is much smaller and it is attached to the β lobe loosely.

δ and ε are subunits specific to the *Firmicutes*. Although the density for δ subunit is weak in σA-RPo, it is strong in σB-RPo (Fig. 1D, F). In agreement with the cryo-EM structures of *Bacillus subtilis*[28,29], the N-terminal domain of δ binds between the β′ subunit shelf and jaw, while the C-terminal region is disordered. Because the C-terminal region is rich in acidic residues, it may prevent the nonspecific interaction between RNAP and DNA[33,34]. The ε subunit is positioned in a cavity formed by the α subunit N-terminal domains, β subunit and β′ subunit, stabilizing the multi-subunit complex.

In the structure of σB-RPo, σB4 clamps the β subunit flap domain tip helix (FTH) and mediates sequence-specific interactions with the promoter −35 element in the same way as σA4 (Fig. 2A and Supplementary Fig. 10). In particular, σB residue R241 is positioned to make a hydrogen bond with the O6 of −35G. The hydroxyl group of σB residue S235 forms a van der Waals interaction with the C5-methyl group of −34T. σB residue M237 makes a van der Waals interaction with N7 of the A opposite −33T. σB residue Q236 is positioned to make a hydrogen bond with N6 of the A opposite −32T. Moreover, there is a potential electrostatic interaction between σB residue R244 and DNA backbone phosphate group. Alanine substitution of these residues does not affect RNAP holoenzyme formation (Supplementary Fig. 11), but hampers σB dependent RPo formation and transcription activity (Fig. 2B, C), confirming that the cryo-EM structure is biologically relevant. In accordance, mutation of the interacting nucleotides impairs σB dependent RPo formation and transcription activity, as well (Fig. 2D, E). Sequence alignment indicates that these residues are divergent between σA and σB (Fig. 2F), explaining the specificity of σB mediated transcription regulation.

In the structure of σA-RPo, σA2 contacts the clamp helices and mediates sequence specific interactions with the promoter −10 element (Fig. 3A). Specifically, σA2 interacts with the first position of the −10 element as dsDNA and the second through sixth positions of the −10 element as ssDNA. σA residue W189 stacks on the base of −12T, forming a wedge that forces the base of −11A to unstack and flip outside the DNA helix, where it is captured by binding within a pocket formed by σA residues F175, K179, F181, and Y186. −7T is flipped out of the base stack and buried deeply in a cognate pocket, as well. In addition, σA1.2 interacts with nontemplate-strand ssDNA extensively, stabilizing the transcription bubble. Especially, residue L111 makes up one wall of the −7T pocket. These interactions are reminiscent of the interaction observed in the crystal structure of *Thermus aquaticus* σA2 in complex with −10 element ssDNA[6].

The most striking feature of σB-RPo is that the upstream 4-bp of the −10 element is co-recognized by σB2 and σB3 as dsDNA (Fig. 3B). The last α helix of σB2 and the first α helix of σB3 bind in the DNA major groove and make sequence specific interactions with the upstream 3 bp of the promoter −10 element. Particularly, σB residues R110 and R100 are positioned to form hydrogen bonds with O6 of −15G and −14G, respectively. σB2 residue R97 is placed to make a van der Waals interaction with N7 of −13G. Strikingly, the fifth bp is unwound and the base of −11A inserts into a hydrophobic pocket formed by σB residues F79 and F86. In addition, σB residue R74 is positioned to form salt bridges with DNA backbone phosphate groups. Alanine substitution of these residues does not affect RNAP holoenzyme formation (Supplementary Fig. 11), but compromises σB dependent RPo formation and transcription activity significantly (Fig. 3C, D), verifying their importance. Consistently, mutation of the interacting nucleotides impairs σB dependent RPo formation and transcription activity, as well (Fig. 3E, F). Again, these residues are divergent between σA and σB (Fig. 3G), explaining the specificity of σB mediated transcription regulation. Although σA1.2 interacts with nontemplate-strand ssDNA extensively in σA-RPo, the density of the nontemplate-strand ssDNA in σB-RPo is weak due to the lack of σB1.2 (Supplementary Fig. 7B).

Previous studies showed that *E. coli* σ70 1.1 modulates the DNA binding activity of σ70. In the absence of RNAP, σ70 1.1 inhibits the DNA binding function of free σ70 [35]. In the presence of RNAP, σ70 1.1 binds in the main channel of RNAP and prevents the nonspecific binding of DNA[36,37]. There is no density for *S. aureus* σA1.1 in the structure of σA-RPo, but the structure of *S. aureus* σA1.1 predicted by AlphaFold is very similar to the structure of *E. coli* σ70 1.1 [36] and *B. subtilis* σA1.1 [38], suggesting their similar roles (Fig. 4A). To delineate the function of *S. aureus* σA1.1, we constructed and purified σA1.1 truncated σA. Fluorescence polarization experiments demonstrate that σA1.1 truncated σA binds promoter DNA better than the full-length σA (Fig. 4B). Moreover, truncation of σA1.1 increases σA dependent RPo formation, confirming that the roles of *S. aureus* σA1.1 and *E. coli* σ70 1.1 are similar (Fig. 4C).

## Discussion

Structural comparison of different σ factors reveals the reason for the short spacers between the −35 element and −10 element of σB regulated promoters. Similar to the structure of *E. coli* σ70-RPo[10], the structure of *S. aureus* σA-RPo demonstrates that σA4 recognizes the promoter −35 element through its helix-turn-helix (HTH) motif and σA2 recognizes the promoter −10 element through two cognate protein pockets (Fig. 5A). Despite the lack of σ1.1, σ1.2, and σ3, the structure of *M. tuberculosis* σH-RPo shows that σH also binds to the promoter in an analogous manner[12]. Like σ70/σA and σH, σB4 recognizes the promoter −35 element through its HTH motif. Unlike σ70/σA and σH, σB2 and σB3 co-recognize the −10 element. Since σ2, σ3, and σ4 are anchored to RNAP surface at the fixed locations, the spacers between the −35 element and −10 element of σB regulated promoters are ~3 bp shorter than those of σ70/σA and σH regulated promoters.

The conversion from RNAP-promoter closed complex (RPc) to RPo has been studied extensively using *E. coli* σ70 [30,31,39]. In σ70-RPc, sequence-specific recognition of the promoter −35 element by σ4 positions the critical and conserved −11A of −10 element in line with σ2 residues that later capture the flipped base to nucleate transcription bubble formation. In σ70-RPo, two conserved pockets in σ70 capture the flipped bases of the −10 element (−11A and −7T) and stabilize the transcription bubble. *S. aureus* σA probably works in the same way as *E. coli* σ70. As for σB, sequence-specific recognition of the promoter −35 element and −10 element by σ4, σ3, and σ2 positions the conserved −11A of −10 element in line with σ2 residues that later capture the flipped base to nucleate transcription bubble formation. Since there is no structural equivalent of the −7T pocket of σ70/σA, only one base of the −10 element (−11A) is flipped and specifically captured in a protein pocket.

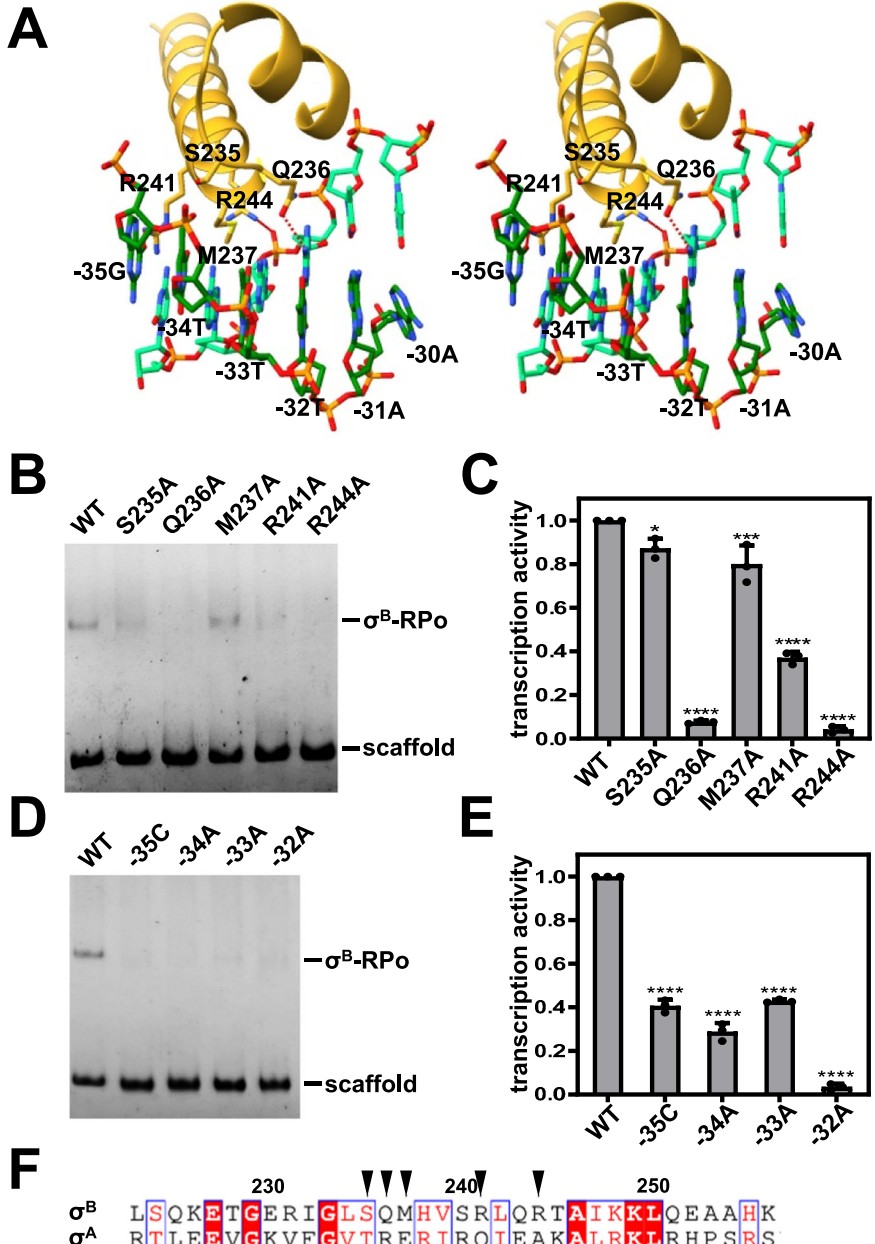

**Fig. 2 | σ-DNA interactions responsible for −35 element recognition. A** σ$^B_4$-DNA interactions are depicted in stereo view. Yellow, σ$^B_4$; dark green, nontemplate strand DNA; light green, template strand DNA. The potential hydrogen bonds are shown as dashed lines. **B** EMSA shows that the substitution of DNA interacting residues impairs σ$^B$-RPo formation. Source data are provided as a Source Data file. **C** Ribogreen transcription assay shows that the substitution of DNA interacting residues impairs σ$^B$ dependent transcription. Error bars represent mean ± SD of $n = 3$ experiments. S235A, $p = 0.0252$; Q236A, $p < 0.0001$; M237A, $p = 0.0008$; R241A, $p < 0.0001$; R244A, $p < 0.0001$. One-way ANOVA. Source data are provided as a Source Data file. **D** EMSA shows that the mutation of the interacting nucleotides impairs σ$^B$-RPo formation. **E** Ribogreen transcription assay shows that the mutation of the interacting nucleotides impairs σ$^B$ dependent transcription. Error bars represent mean ± SD of $n = 3$ experiments. −35C, $p < 0.0001$; −34A, $p < 0.0001$; −33A, $p < 0.0001$; −32A, $p < 0.0001$. One-way ANOVA. Source data are provided as a Source Data file. **F** Sequence alignment of *S. aureus* σ$^A$ and σ$^B$. The DNA interacting residues of σ$^B$ are indicated by black triangles. Source data are provided as a Source Data file.

σ$^B$ orthologs are presented in many Gram-positive bacteria, such as *Bacillus subtilis*, *Bacillus cereus*, *Clostridium difficile*, *Listeria monocytogenes*, and *Mycobacterium tuberculosis* (σ$^F$, instead of σ$^B$, is a σ$^B$ ortholog). Their regulated promoters all share similar consensus sequences and spacer lengths[21,40–44]. Sequence alignment indicates that the DNA interacting residues identified in this work are highly conserved among these σ$^B$ orthologs (Fig. 5B), suggesting the finding of this work can be applied to other bacteria, as well.

Our cryo-EM structures also hints at mechanisms of action for δ subunit during transcription initiation. Our cryo-EM structures

demonstrate that the N-terminal domain of δ subunit anchors the C-terminal region at the rim of the main channel, where the C-terminal region can reach into the main channel and exclude the binding of DNA (Supplementary Fig. 8). Accordingly, δ subunit has been shown to interfere with the interaction between DNA and RNAP[33,34]. Our biochemical experiments indicate that σ$^A_{1.1}$ also resides in the main channel and exclude the binding of DNA. Therefore, the C-terminal region of δ subunit probably competes with σ$^A_{1.1}$ and occupies similar regions in the main channel. Consistently, *Bacillus subtilis* δ subunit exhibits negative cooperativity

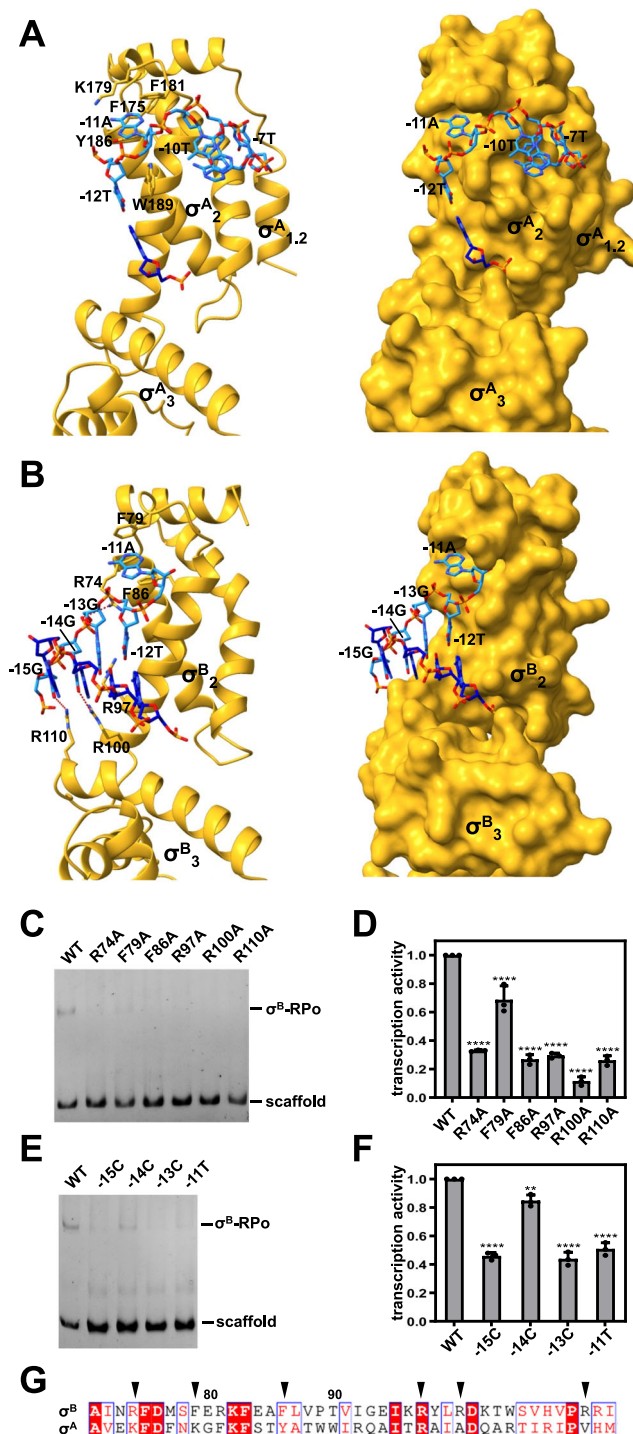

**Fig. 3 | σ-DNA interactions responsible for −10 element recognition. A** $\sigma^A_2$-DNA interactions. Yellow, σ; light blue, nontemplate strand DNA; dark blue, template strand DNA. Left subpanel, ribbon representation; right subpanel, surface representation. **B** $\sigma^B_2$ and $\sigma^B_3$ co-recognize the −10 element. Yellow, σ; light blue, nontemplate strand DNA; dark blue, template strand DNA. Left subpanel, ribbon representation; right subpanel, surface representation. **C** EMSA shows that the substitution of DNA interacting residues impairs $\sigma^B$-RPo formation. Source data are provided as a Source Data file. **D** Ribogreen transcription assay shows that the substitution of DNA interacting residues impairs $\sigma^B$ dependent transcription. Error bars represent mean ± SD of $n = 3$ experiments. R74A, $p < 0.0001$; F79A, $p < 0.0001$; F86A, $p < 0.0001$; R97A, $p < 0.0001$; R100A, $p < 0.0001$; R110A, $p < 0.0001$. One-way ANOVA. Source data are provided as a Source Data file. **E** EMSA shows that the mutation of the interacting nucleotides impairs $\sigma^B$-RPo formation. **F** Ribogreen transcription assay shows that the mutation of the interacting nucleotides impairs $\sigma^B$ dependent transcription. Error bars represent mean ± SD of $n = 3$ experiments. −15C, $p < 0.0001$; −14C, $p = 0.0030$; −13C, $p < 0.0001$; −11T, $p < 0.0001$. One-way ANOVA. Source data are provided as a Source Data file. **G** Sequence alignment of *S. aureus* $\sigma^A$ and $\sigma^B$. The DNA interacting residues of $\sigma^B$ are indicated by black triangles.

## Methods

### Expression and purification of *S. aureus* RNAP core enzyme

Genes *rpoA*, *rpoB*, *rpoC*, *rpoZ*, *rpoE*, and *rpoY* were amplified from *S. aureus* strain N315 and subcloned into the pET21a vector by homologous recombination (Supplementary Data 1). 10 x histidine codons were placed after rpoC gene to facilitate purification. *S. aureus* RNAP core enzyme was prepared from *E. coli* strain BL21(DE3) (Invitrogen, Inc.) transformed with plasmid pET21a-Sau-rpoABCZEY. Single colonies of the resulting transformants were used to inoculate 100 mL LB broth containing 100 μg/mL ampicillin, and cultures were incubated 16 h at 37 °C with shaking. Aliquots (10 mL) were used to inoculate 1 L LB broth containing 100 μg/mL ampicillin, cultures were incubated at 37 °C with shaking until $OD_{600}$ = 0.6, cultures were induced by addition of IPTG to 0.5 mM, and cultures were incubated 15 h at 20 °C. Then cells were harvested by centrifugation (5000×$g$; 15 min at 4 °C), resuspended in 30 mL lysis buffer (20 mM Tris-HCl, pH 8.0, 0.5 M NaCl, 2 mM EDTA, 5% glycerol, and 5 mM DTT) and lysed using a JN-02C cell disrupter (JNBIO, Inc.). After poly(ethyleneimine) precipitation and ammonium sulfate precipitation, the pellet was resuspended in buffer A (20 mM Tris-HCl, pH 8.0, 0.5 M NaCl, and 5% glycerol) and loaded onto a 5 mL column of Ni-NTA agarose (Qiagen, Inc.) equilibrated with buffer A. The column was washed with 25 mL buffer A containing 20 mM imidazole and eluted with 25 mL buffer A containing 0.3 M imidazole. The eluate was further purified by anion-exchange chromatography on a HiTrap Q HP column (GE Healthcare, Inc.). Fractions containing *S. aureus* RNAP core enzyme were applied to a HiLoad 16/600 Superdex 200 column (GE Healthcare, Inc.) equilibrated in 10 mM HEPES, pH 7.5, and 100 mM KCl, and the column was eluted with 120 mL of the same buffer. Fractions containing *E. coli* RNAP core enzyme were stored at −80 °C. Yield was ~0.6 mg/L.

### Expression and purification of *S. aureus* $\sigma^A$ and $\sigma^B$

Genes encoding *S. aureus* $\sigma^A$ and $\sigma^B$ were amplified from *S. aureus* strain N315 and subcloned into pET21a vector. *S. aureus* $\sigma^A$ and $\sigma^B$ were prepared from *E. coli* strain BL21(DE3) (Invitrogen, Inc.) transformed with plasmids pET21a-$\sigma^A$ and pET21a-$\sigma^B$, respectively. Single colonies of the resulting transformants were used to inoculate 1L LB broth containing 100 μg/mL ampicillin, cultures were incubated at 37 °C with shaking until $OD_{600}$ = 0.6, cultures were induced by addition of IPTG to 0.5 mM, and cultures were incubated 15 h at 20 °C. Then cells were harvested by centrifugation (5000×$g$; 15 min at 4 °C), resuspended in 30 mL buffer B (20 mM Tris-HCl, pH 8.0, 0.5 M NaCl) and lysed using a JN-02C cell disrupter (JNBIO, Inc.). The lysate was centrifuged (20,000×$g$; 45 min at 4 °C), and the supernatant was loaded onto a 5 mL column of Ni-NTA agarose (Qiagen, Inc.) equilibrated with

with $\sigma^A$ and favors its exchange for alternative σ factors that lack $\sigma^A_{1.1}$[45–47].

RNAP inhibitor rifampin has been successful in treating *S. aureus* infection, especially periprosthetic joint infection[48]. However, the danger of rapid emergence of resistance restricts its usage[49]. Resistance to rifampin in *S. aureus* is determined by mutations in the gene encoding the RNAP β subunit. The structures presented here provide a structural basis for these resistant mutations. After we dock the rifampin into our structures (Supplementary Fig. 12), we find that all resistant mutations are positioned within 10 Å from rifampin. Some of them even directly contact rifampin. For example, substitution of β residue H481 would be expected to disrupt two hydrogen bonds between RNAP and rifampin.

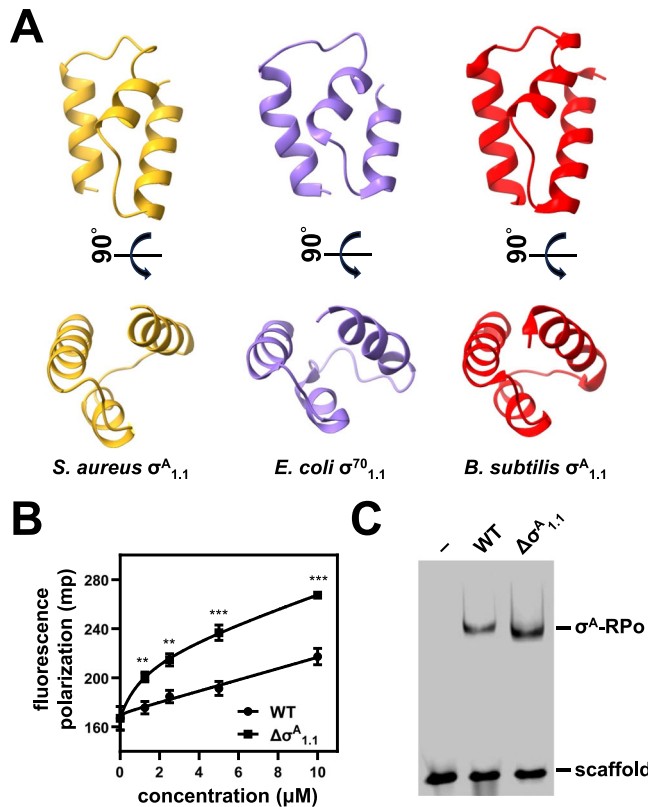

**Fig. 4 | *S. aureus* σ$^A_{1.1}$ suppresses the DNA binding activity of free σ$^A$ and σ$^A$-RNAP holoenzyme. A** Structural comparison of *S. aureus* σ$^A_{1.1}$, *E. coli* σ$^{70}_{1.1}$ (PDB: 4LK1), and *B. subtilis* σ$^A_{1.1}$ (PDB: 5MWW). The structure of *S. aureus* σ$^A_{1.1}$ is predicted by AlphaFold. **B** Fluorescence polarization assay shows that σ$^A_{1.1}$ truncated σ$^A$ binds promoter DNA better than the full-length σ$^A$. Error bars represent mean ± SD of $n = 3$ experiments. 1.25 μM, $p = 0.0027$; 2.5 μM, $p = 0.0019$; 5 μM, $p = 0.0008$; 10 μM, $p = 0.0003$. Two-tailed Student's $t$ test. Source data are provided as a Source Data file. **C** EMSA shows that truncation of σ$^A_{1.1}$ increases σ$^A$-dependent RPo formation. Source data are provided as a Source Data file.

buffer B. The column was washed with 25 mL buffer B containing 20 mM imidazole and eluted with 25 mL buffer B containing 0.3 M imidazole. The eluate was further purified by anion-exchange chromatography on a HiTrap Q HP column (GE Healthcare, Inc.). Fractions containing *S. aureus* σ$^A$ and σ$^B$ were applied to a HiLoad 16/600 Superdex 200 column (GE Healthcare, Inc.) equilibrated in 10 mM HEPES, pH 7.5, and 100 mM KCl, and the column was eluted with 120 mL of the same buffer. Fractions containing *S. aureus* σ$^A$ and σ$^B$ were stored at −80 °C. Yields were ~3 mg/L. Mutant proteins were purified in the same way as wild-type protein.

### Expression and purification of *S. aureus* RNAP holoenzyme
*S. aureus* RNAP core enzyme and *S. aureus* σ were incubated in a 1:4 ratio for 1 h at 4 °C. The reaction mixtures were applied to a Superose 6 column (GE Healthcare, Inc.) equilibrated in 10 mM HEPES, pH 7.5, and 100 mM KCl, and the column was eluted with 24 mL of the same buffer. Fractions containing *S. aureus* RNAP holoenzyme were stored at −80 °C.

### Primer extension transcription assay
5′ 6-FAM labeled RNA and template strand DNA (Supplementary Data 1) were annealed at a 1:1 ratio in 50 mM Tris-HCl, pH 8.0, 0.1 M KCl, and 10 mM MgCl$_2$. Primer extension transcription assay was performed in reaction mixtures (20 μl) containing 1.2 μM hybrid, 1 μM *S. aureus* RNAP, 50 mM Tris-HCl, pH 8.0, 0.1 M KCl, and 10 mM MgCl$_2$. Reaction mixtures were incubated for 15 min at 25 °C, supplemented

with 1.3 μM nontemplate strand DNA. After 15 min at 25 °C, 1 mM ATP and 1 mM GTP were added. Primer extension was allowed to proceed for 15 min at 37 °C. Reactions were terminated by adding 20 μl loading buffer (10 mM EDTA, 0.02% bromophenol blue, 0.02% xylene cyanol, and 8 M urea) and boiling for 2 min. Products were applied to 15% urea-polyacrylamide slab gels (19:1 acrylamide/bisacrylamide), electrophoresed in 90 mM Tris-borate (pH 8.0) and 0.2 mM EDTA, and analyzed by Typhoon (GE Healthcare, Inc.).

### Run-off transcription assay
Nontemplate strand DNA and template strand DNA (Supplementary Data 1) were annealed at a 1:1 ratio in 50 mM Tris-HCl, pH 8.0, 0.1 M KCl, and 10 mM MgCl$_2$. Run-off transcription assay was performed in reaction mixtures (10 μl) containing 5 nM DNA, 100 nM *S. aureus* RNAP holoenzyme, 50 mM Tris-HCl, pH 8.0, 0.1 M KCl, and 10 mM MgCl$_2$. Reaction mixtures were incubated for 10 min at 37 °C, supplemented with 0.2 mM CTP, 0.2 mM UTP, 0.2 mM GTP, and 0.2 μl 3.3 μM [α-$^{32}$P] ATP (100 Bq/fmol). RNA synthesis was allowed to proceed for 10 min at 37 °C. Reactions were terminated by adding 10 μl loading buffer (10 mM EDTA, 0.02% bromophenol blue, 0.02% xylene cyanol, and 8 M urea) and boiling for 2 min. Products were applied to 15% urea-polyacrylamide slab gels (19:1 acrylamide/bisacrylamide), electrophoresed in 90 mM Tris-borate (pH 8.0) and 0.2 mM EDTA, and analyzed by storage-phosphor scanning.

### Ribogreen transcription assay
DNA fragments corresponding to −55 to −1 of *S. aureus* rrnB P1 and yabJ promoters followed by 311 bp random sequence and tR2 terminator (Supplementary Data 1) were synthesized and inserted into a pUC vector (Genewiz, Inc.). The DNA fragments were amplified by PCR and purified using the QIAquick PCR Purification Kit (Qiagen, Inc.). Ribogreen transcription assay was performed in 96-well flat-bottom black microplates. Reaction mixtures (20 μL) contained 20 nM DNA, 100 nM *S. aureus* RNAP holoenzyme, 1 mM NTPs, 50 mM Tris-HCl, pH 8.0, 0.1 M KCl, and 10 mM MgCl$_2$. Reaction mixtures were incubated for 60 min at 37 °C, supplemented with 1 μL of 5 mM CaCl$_2$ and 1 μL of DNase I (ThermoFisher, Inc.). DNA digestion was allowed to proceed for 90 min at 37 °C. Reactions were terminated by adding 1:500 diluted ribogreen (Invitrogen, Inc.) in 100 μL TE buffer (10 mM Tris-HCl, pH 8.0, 1 mM EDTA). Fluorescence emission intensities were measured using a Varioskan Flash Multimode Reader (ThermoFisher, Inc.; excitation wavelength = 485 nm; emission wavelength = 528 nm).

### Fluorescence polarization assay
3′ 6-FAM labeled template strand DNA and unmodified nontemplate strand DNA (Supplementary Data 1) were annealed at a 1:1 ratio in 50 mM Tris-HCl, pH 8.0, 0.1 M KCl, and 10 mM MgCl$_2$. Equilibrium fluorescence polarization assays were performed in a 96-well microplate format. Reaction mixtures (100 μl) contained: 0–10 μM σ$^A$ or σ$^A$ derivative, 0.1 μM 6-FAM-labeled DNA scaffold, 50 mM Tris-HCl, pH 8.0, 0.1 M KCl, and 10 mM MgCl$_2$. Following incubation mixtures for 10 min at 25 °C, fluorescence emission intensities were measured using a BioTek Synergy H1 microplate reader (Agilent, Inc.; excitation wavelength = 485 nm; emission wavelength = 525 nm). Fluorescence polarization was calculated using:

$$P = (I_{VV} - I_{VH})/(I_{VV} + I_{VH}) \tag{1}$$

where $I_{VV}$ and $I_{VH}$ are fluorescence intensities with the excitation polarizer at the vertical position and the emission polarizer at, respectively, the vertical position and the horizontal position.

### Electrophoretic mobility shift assay
Electrophoretic mobility shift assay was performed using the same DNA fragments as ribogreen transcription assay. Reaction mixtures

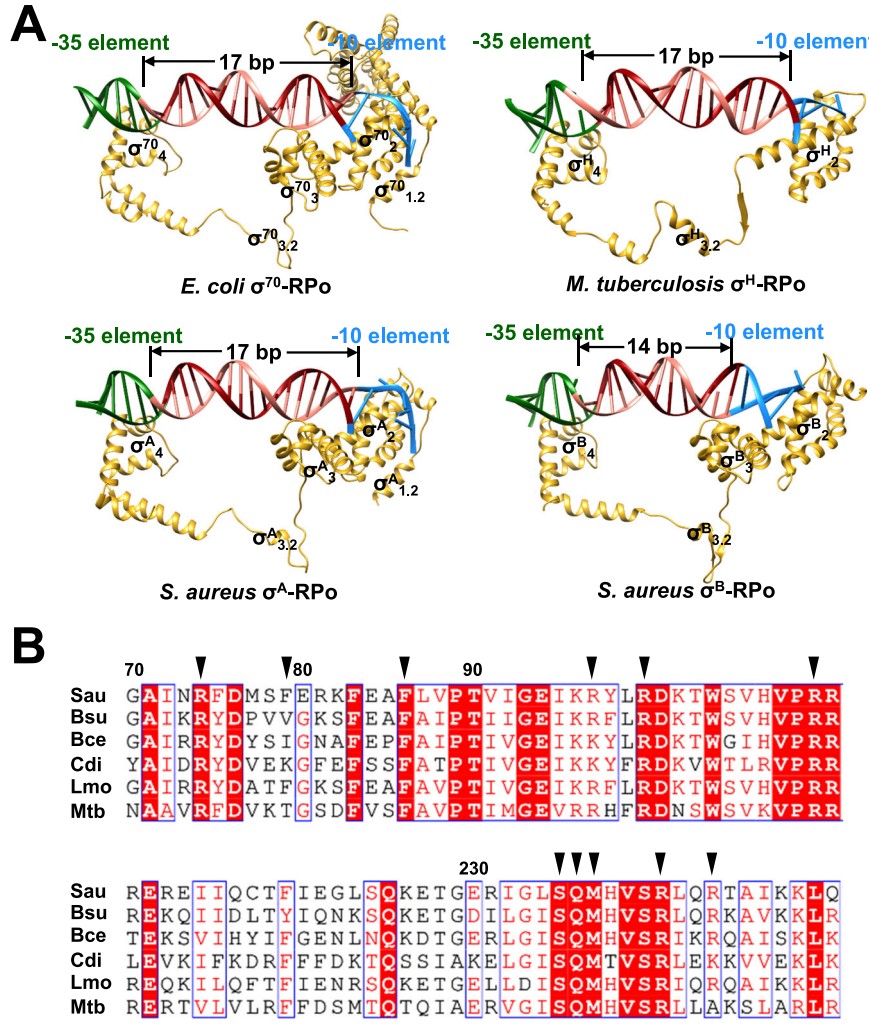

**Fig. 5 | Structural comparison of *E. coli* σ⁷⁰-RPo, *M. tuberculosis* σᴴ-RPo, *S. aureus* σᴬ-RPo, and σᴮ-RPo. A** σ-DNA interactions in *E. coli* σ⁷⁰-RPo (PDB: 6CA0), *M. tuberculosis* σᴴ-RPo (PDB: 5ZX2), *S. aureus* σᴬ-RPo, and σᴮ-RPo. Yellow, σ; salmon, nontemplate strand DNA; red, template strand DNA; green, −35 element; blue, −10 element. **B** Sequence alignment of σᴮ orthologs from *S. aureus* (Sau), *Bacillus subtilis* (Bsu), *Bacillus cereus* (Bce), *Clostridium difficile* (Cdi), *Listeria monocytogenes* (Lmo), and *Mycobacterium tuberculosis* (Mtb). The DNA interacting residues of *S. aureus* σᴮ are indicated by black triangles.

(20 µL) contained 40 nM DNA, 100 nM *S. aureus* RNAP holoenzyme, 50 mM Tris-HCl, pH 8.0, 0.1 M KCl, and 10 mM MgCl₂. Reaction mixtures were incubated for 10 min at 37 °C. The reaction mixtures were applied to 5% polyacrylamide slab gels (29:1 acrylamide/bisacrylamide), electrophoresed in 90 mM Tris-borate, pH 8.0, and 0.2 mM EDTA, stained with 4 S Red Plus Nucleic Acid Stain (Sangon Biotech, Inc.).

**Cryo-EM grid preparation**
Template strand DNA and non-template strand DNA (Genewiz, Inc.) were annealed at a 1:1 ratio in 10 mM HEPES, pH 7.5, 0.1 M KCl. Reaction mixtures (20 µL) contained 1.2 µM DNA scaffold, 1 µM *S. aureus* RNAP holoenzyme, 10 mM HEPES, pH 7.5, 0.1 M KCl. Reaction mixtures were incubated for 10 min at 37 °C. Quantifoil grids (R 1.2/1.3, Cu, 300) were glow-discharged for 120 s at 25 mA prior to the application of 3 µL of the samples, then plunge-frozen in liquid ethane using a Vitrobot (FEI, Inc.) with 95% chamber humidity at 10 °C.

**Cryo-EM data acquisition and processing**
The grids were imaged using a 300 kV Titan Krios equipped with a Falcon 4 direct electron detector (FEI, Inc.). Images were recorded with EPU in counting mode with a physical pixel size of 1.19 Å and a defocus range of 1.0-2.0 µm. Images were recorded with a 7.36 s

exposure to give a total dose of 51 e/Å². Subframes were aligned and summed using RELION's own implementation of the UCSF MotionCor2[50]. The contrast transfer function was estimated for each summed image using CTFFIND4[51]. From the summed images, approximately 1000 particles were manually picked and subjected to 2D classification in RELION[52]. 2D averages of the best classes were used as templates for auto-picking in RELION. Auto-picked particles were manually inspected, then subjected to 2D classification in RELION. Poorly populated classes were removed. The remaining particles were 3D classified in RELION using a map of *E. coli* TEC (EMD-8585 [https://www.ebi.ac.uk/pdbe/entry/emdb/EMD-8585])[53] low-pass filtered to 40 Å resolution as a ref. 3.D classification resulted in 4 classes, among which only one class has a clear density for RNAP. Particles in this class were 3D auto-refined. CTF refinement and particle polishing were performed before final 3D refinement and postprocessing.

**Cryo-EM model building and refinement**
The models of *S. aureus* α, β, β′, ω, δ, ε, σᴬ, and σᴮ predicted by AlphaFold[54] were fitted into the cryo-EM density map using Chimera[55] and were adjusted in Coot[56]. The models of DNA scaffolds were built manually in Coot. The coordinates were real-space refined with secondary structure restraints in Phenix[57].

## Statistics and reproducibility

Statistics were performed in GraphPad Prism 8.0.2. No statistical method was used to predetermine sample size. No data were excluded from the analyses. The experiments were not randomized.

## Reporting summary

Further information on research design is available in the Nature Portfolio Reporting Summary linked to this article.

## Data availability

The cryo-EM density maps generated in this study have been deposited in the Electron Microscopy Data Bank under accession codes EMD-38087 and EMD-38088. The atomic models generated in this study have been deposited in the Protein Data Bank under accession codes 8X6F and 8X6G. The cryo-EM density map used in this study is available in the Electron Microscopy Data Bank under accession code EMD-8585. The atomic models used in this study are available in the Protein Data Bank under accession codes 4LK1, 5MWW, 5ZX2, and 6CA0. Source data are provided with this paper.

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

## Acknowledgements

We thank Shenghai Chang at the Center of Cryo Electron Microscopy in Zhejiang University School of Medicine and Liangliang Kong at the cryo-EM center of the National Center for Protein Science Shanghai for help with cryo-EM data collection. We thank Cheng Ma and Li Liu from the Core Facilities, Zhejiang University School of Medicine for their technical support. This work was funded by National Key R&D Program of China (2023YFC2307100 to Y.F.), National Natural Science Foundation of China (32270030 to Y.F.).

## Author contributions

L.Y., Q.L., L.X., and B.W. performed the experiments. Y.F. supervised the experiments. All authors contributed to the analysis of the data and the interpretation of the results. Y.F. wrote the manuscript with contributions from the other authors.

## Competing interests

The authors declare no competing interests.
