## [Peer Review File · Nature Communications]

Structural basis of promoter recognition by Staphylococcus aureus RNA polymeraseREVIEWER COMMENTS

Reviewer #1 (Remarks to the Author):

This is a short and sweet manuscript that describes an interesting result regarding *S. aureus* RNA polymerase (RNAP, or E) sigmaB-holoenzyme (EsigmaB) recognition of its divergent promoter sequences. EsigmaB controls the expression of genes important for virulence, etc. The consensus promoter for the housekeeping sigma (sigmaA) is TTGACA(17bp spacer)TATAAT (i.e. the -35 and -10 elements are separated by an optimal 17 bp spacer. However, sigmaB has an unusual consensus; GTTTWW(14 bp spacer)GGGWAW. In the structures of the promoter complexes, in essence the 'A' of the sigmaB consensus aligns with the 5'-A of the sigmaA consensus (TATAAT). The 'GGG' of the sigmaB -10 motif remains double-stranded but is recognized through the major groove by residues of sigmaB. These results are interesting but some issues with the manuscript need to be addressed before publishing:

Major issues

1. Results, 2nd to last para.: Here the authors describe the recognition of the sigmaA -10 element by EsigmaA. Many equivalent structures have been observed previously, the first being Feklistov et al., 2011 (pmid: 22136875). This previous work needs to be cited here.
2. Results: the manuscript describes the unusual recognition of the -10 element by the EsigmaB, but does not mention the significant fact that an equivalent of the non-template strand -7T (a conserved feature of EsigmaA promoter -10 element recognition) is not a part of the sigmaB consensus. In EsigmaA, the -7T is flipped out of the single-stranded DNA base stack and bound in a cognate pocket of sigmaA (Feklistov et al., 2011; pmid: 22136875). The single-stranded DNA downstream of the sigmaB -10 element is disordered in the EsigmaB-RPo (Figure 3B) so an equivalent interaction apparently does not occur. However, is a pocket similar to the -7T recognition pocket of sigmaA present in the structure of sigmaB? If a 'T' was present in the non-template strand of a sigmaB promoter (i.e. GGGWAWNNT...), would it be recognized?
3. In several of the Figures (Figure 1, 3, and 4), a bright yellow color was chosen to represent sigma. Please use a different color, the figures with yellow are very 'washed out' and difficult to see.
4. In Figure 2A, details of sigmaA4 and sigmaB4 interactions with their respective -35 elements are shown. These should be accompanied (either in the main Figures or in Supplement Figures) with density maps that show that these interactions can actually be discerned in the cryo-EM density. The authors use mutagenesis of sigmaB to show that the observed interactions are important for promoter interaction (Figures 2B and 2C), but the authors need to demonstrate that the mutant sigmaB's can interact with core RNAP and form holoenzyme.
5. In Figures 3A and 3B, details of sigmaA2 and sigmaB2 interactions with their respective -10 elements are shown. These should be accompanied (either in the main Figures or in Supplement Figures) with density maps that show that these interactions can actually be discerned in the cryo-EM density. The authors use mutagenesis of sigmaB to show that the observed interactions are important for promoter interaction (Figures 3C and 3D), but the authors need to demonstrate that the mutant sigmaB's can interact with core RNAP and form holoenzyme.
6. I'm not really sure that Figure 4A adds much to the manuscript.
7. In the experiments of Figure 4B, rather than use heparin to assess promoter stability, it would be better to use promoter lifetime assays (Ross & Gourse, 2009; pmid: 18952176) with promoter DNA as a sink (for example, see Davis et al., 2015; pmid: 25510492).
8. Figures S4 and S6 - please present 3DFSCs (<https://3dfsc.salk.edu/>) to ensure that the maps are not distorted by the uneven particle orientation distribution.

Minor issues

9. It would be informative to include sequence logos (see Shultzaberger et al., 2007; pmid: 1718927) for the sigmaB promoter consensus elements.

Reviewer #2 (Remarks to the Author):

The manuscript by Yuan et al. presents the first high-resolution structure of the RNA polymerase from *Staphylococcus aureus*, a significant human pathogen. By expressing recombinant polymerase subunits in *E. coli* cells, the authors successfully reconstituted both core and holo SAU RNAP. This achievement represents an important milestone in SAU RNAP research. The structural data obtained for holoenzyme assembled with different sigma factors (sigma A and B) and promoter DNA, combined with functional assays and mutagenesis, elucidate the mechanism behind the distinct promoter specificities of these transcription factors. The structures also pave the way for structure-assisted studies on SAU RNAP, potentially leading to the development of new antibiotics and thus constitutes a significant contribution to the field. However, the structural findings obtained in this study are not sufficiently analyzed, and the level of insights on the structural basis of promoter recognition is insufficient. The manuscript would further benefit from a detailed analysis of the overall architecture of SAU RNAP and a comparison of the obtained complexes with the available structures of initiation complexes of other well-studied bacterial RNAP. The presentation of the figures and writing should also be improved. A major revision of the manuscript is required to address the issues summarized below:

1. While the authors claim that the obtained recombinant RNAP is active and suitable for structural studies, additional data are needed for justification.

First, the cloning of core SAU RNAP is not described in sufficient detail. Please include the exact pET vector description, and present a map of the obtained plasmid and its complete sequence.

Second, the activity of the core enzyme is not demonstrated. This activity must be measured using a primer extension assay on an RNA-DNA scaffold.

Finally, the activity of the holoenzyme is not demonstrated sufficiently because the transcription assay does not distinguish between run-off and abortive products. It is, therefore, impossible to determine whether the recombinant holoenzyme is capable of productive transcription or if it can only form off-path complexes on promoter DNA.

2. To better evaluate the quality of the model fitting, the authors should provide map-to-model Fourier Shell Correlation (FSC) plots, highlighting resolution cutoffs at FSC = 0.5. The map-to-model correlation coefficients (CC) should be added to the Table S1.

3. A supporting figure showing the sigma R2/R3 -10 DNA interface from the Rpo-sigma B structure should be provided with the segment of the cryoEM density map. Residues critical for -10 region recognition, such as R74, R100, and R110, should be highlighted.

4. The rotamer outliers and clash scores should be improved for PDB 8X6F. The latest versions of Phenix can improve the rotamer outliers.

5. The manuscript should discuss the overall structure of SAU RNAP, highlighting differences from the previously published structures of bacterial polymerases.

6. Introduce a figure comparing interactions of the -35 promoter region with Sigma A and B.

7. Comparison to available structures of bacterial transcription initiation complexes (e.g. Sigma70/SigA in *E. coli*, and Sigma H in *M. tuberculosis*) should be included in the manuscript. This would help readers understand different modes of sigma factor binding to -10/-35 promoter DNA.

8. The manuscript will benefit from a more specific title. Perhaps " Structural basis of promoter recognition by Staphylococcus aureus RNAP" will do the job.
9. The significance of delta and epsilon subunits is not sufficiently discussed.
10. There are no references to Fig 1A and Fig1B in the text of the manuscript.
11. Figure 3A lacks Sigma A region 3, making comparison with Sigma B difficult.
12. Regions -35 and -10 are highlighted in green and blue in Fig 1B. Keeping these colors consistent with the other figures in the manuscript will be helpful.
13. Statements about the clamp are not illustrated in the manuscript.
14. The manuscript will benefit from professional editing to improve grammar and eliminate jargon : Fig3 legend: change "jeopardizes" to "impairs", "Expressed recombinantly" - to expression of recombinant protein, "Excessive sigma factor" to "an excess of a sigma factor", etc

Reviewer #3 (Remarks to the Author):

The manuscript by Yuan et al. describes two cryo-EM structures of Staphylococcus aureus RNA polymerase in complex with its two sigma factors (sigmaA and sigmaB) and respective promoters. sigmaA is responsible for transcribing house-keeping genes and sigmaB virulence factors. The quality of the structures is difficult to assess as no detailed electron density is shown or described adequately. The manuscript is poorly written and does not provide any mechanistic insights into the action and gene regulation by the two sigma factors. The DNA structures are also poorly built and the models look distorted in figures.

Main points:

- 1) The Introduction is superficial. It explains extensively the importance of S. aureus and sigmaB on virulence but the results here do not explain how that is achieved so the content is unnecessarily detailed.
- 2) Instead, there is no information on the two sigma factors. What is known biochemically ? How are they regulated ? Do they need to be activated ? How do they convert from closed to open complex ?
- 3) In addition, functional similarities to other systems including other gram-positive and gram-negative sigma70, ECF sigma and sigma54 should be discussed including consensus sequences and the spacers between the two recognition sites.
- 4) Structures of RNAP and sigma factors and what are known about them functionally should be explained.
- 5) There are no descriptions or discussions on the RNAP and sigma factors. What are the structural similarities and differences with other known sigma factors and their interactions with DNA ?
- 6) The main differences between the two structures lie in the fact that in sB, region 2 and region 3 of sigma are involved in DNA binding, they argue this causes/explains the shorter spacer. Which is the major DNA binding site ? Does Region 4 binds to -35 first, what is the order of binding ?
- 7) What is the role of region 1 in sigmaA ? Are they similar to sigma70 ? What are the similarities and differences
- 8) Can sigmaA and sigmaB bind to DNA in the absence of RNAP ?
- 9) Do they see density for transcription bubble ? what causes the conversion from closed to open complex in their sample preparation ?

Other points:

- 1) Fig. S1, lanes containing sB and RNAP core samples have multiple other bands. What are they ?

- 2) Detailed electron density should be shown for DNA and regions of protein-DNA interactions to provide confidence in the quality of the reconstructions.
- 3) FSC for masked, unmasked and phase randomised should also be shown.
- 4) Figures are overall poor. Figures should show different regions of sigma regions, accomplished by density in regions that show detailed interactions.
- 5) Fig. 2, B and D, for WT, why are the two EMSA result different ? Same is true for Fig. 3 C and E.
- 6) Fig. 4B, sigmaB-RPo is less stable to start with. This can't be explained by the lack of region 1 as in sigmaB has additional interactions with region 3.

Responses to Reviewers' Comments

Reviewer #1:

This is a short and sweet manuscript that describes an interesting result regarding *S. aureus* RNA polymerase (RNAP, or E) sigmaB-holoenzyme (EsigmaB) recognition of its divergent promoter sequences. EsigmaB controls the expression of genes important for virulence, etc. The consensus promoter for the housekeeping sigma (sigmaA) is TTGACA(17bp spacer)TATAAT (i.e. the -35 and -10 elements are separated by an optimal 17 bp spacer. However, sigmaB has an unusual consensus; GTTTWW(14 bp spacer)GGGWAW. In the structures of the promoter complexes, in essence the 'A' of the sigmaB consensus aligns with the 5'-A of the sigmaA consensus (TATAAT). The 'GGG' of the sigmaB -10 motif remains double-stranded but is recognized through the major groove by residues of sigmaB. These results are interesting but some issues with the manuscript need to be addressed before publishing:

Major issues

1. Results, 2nd to last para.: Here the authors describe the recognition of the sigmaA -10 element by EsigmaA. Many equivalent structures have been observed previously, the first being Feklistov et al., 2011 (pmid: 22136875). This previous work needs to be cited here.

We would like to thank the reviewer for his/her time in reviewing our manuscript and also for the detailed instructions to improve the quality of the manuscript. The reference has been described and cited on page 5.

2. Results: the manuscript describes the unusual recognition of the -10 element by the EsigmaB, but does not mention the significant fact that an equivalent of the non-template strand -7T (a conserved feature of EsigmaA promoter -10 element recognition) is not a part of the sigmaB consensus. In EsigmaA, the -7T is flipped out of the single-stranded DNA base stack and bound in a cognate pocket of sigmaA (Feklistov et al., 2011; pmid: 22136875). The single-stranded DNA downstream of the sigmaB -10 element is disordered in the EsigmaB-RPo (Figure 3B) so an equivalent interaction apparently does not occur. However, is a pocket similar to the -7T recognition pocket of sigmaA present in the structure of sigmaB? If a 'T' was present in the non-template strand of a sigmaB promoter (i.e. GGGWAWNNT...), would it be recognized?

According to the structure of σ^A -RPo, residue L111 of $\sigma^A_{1,2}$ makes up one wall of the -7T pocket. Since there is no $\sigma^B_{1,2}$ in σ^B , the -7T pocket is absent in σ^B , as well. The difference has been described on page 7.

3. In several of the Figures (Figure 1, 3, and 4), a bright yellow color was chosen to represent sigma. Please use a different color, the figures with yellow are very 'washed out' and difficult to see.

Bright yellow has been changed to dark yellow in the revised manuscript.

4. In Figure 2A, details of sigmaA4 and sigmaB4 interactions with their respective -35 elements are shown. These should be accompanied (either in the main Figures or in Supplement Figures) with density maps that show that these interactions can actually be discerned in the cryo-EM density. The authors use mutagenesis of sigmaB to show that the observed interactions are important for promoter interaction (Figures 2B and 2C), but the authors need to demonstrate that the mutant sigmaB's can interact with core RNAP and form holoenzyme.

Density maps showing these interactions have been added in Figure S7. Mutant σ^B was incubated with RNAP core enzyme. Then size exclusion chromatography and SDS-PAGE were used to confirm that mutant σ^B can form holoenzyme with RNAP core. The image of SDS-PAGE have been added in Figure S11. The added information is excerpted below:

Supplementary Figure 7. Representative electron potential map and superimposed model.

(A) The electron potential map without B-factor sharpening and the superimposed model of σ^A and σ^B . The contour level is 0.01. The carve radius is 3.4 Å.

(B) The electron potential map without B-factor sharpening and the superimposed model of *rrnB* P1 and *yabJ*. The contour level is 0.01. The carve radius is 3.4 Å.

(C) The electron potential map with B-factor sharpening and the superimposed model of σ^B -DNA. The contour level is 0.01. The carve radius is 3.4 Å.

(D) The electron potential map with B-factor sharpening and the superimposed model of σ^A -DNA and σ^B -DNA. The contour level is 0.01. The carve radius is 3.4 Å.

Supplementary Figure 11. σ^B derivatives were incubated with RNAP core enzyme. Then size exclusion chromatography and SDS-PAGE were used to confirm that σ^B derivatives can form holoenzyme with RNAP core.

5. In Figures 3A and 3B, details of sigmaA2 and sigmaB2 interactions with their respective -10 elements are shown. These should be accompanied (either in the main Figures or in Supplement Figures) with density maps that show that these interactions can actually be discerned in the cryo-EM density. The authors use mutagenesis of sigmaB to show that the observed interactions are important for promoter interaction (Figures 3C and 3D), but the authors need to demonstrate that the mutant sigmaB's can interact with core RNAP and form holoenzyme.

Density maps showing these interactions have been added in Figure S7. Mutant σ^B was incubated with RNAP core enzyme. Size exclusion chromatography and SDS-PAGE were used to confirm that mutant σ^B can form holoenzyme with RNAP core. The image of SDS-PAGE have been added in Figure S11.

6. I'm not really sure that Figure 4A adds much to the manuscript.

To make Figure 4A (Figure 5A in the revised manuscript) more informative, structures of σ^{70} -RPO and σ^H -RPO have been added for comparison.

7. In the experiments of Figure 4B, rather than use heparin to assess promoter stability, it would be better to use promoter lifetime assays (Ross & Gourse, 2009; pmid: 18952176) with promoter DNA as a sink (for example, see Davis et al., 2015; pmid: 25510492).

Lifetime assay has been performed using competitive promoter DNA with a consensus -35 element, a consensus -10 element and a transcription bubble maintained in the unwound state by having noncomplementary sequences on nontemplate and template strands. Unexpectedly, both σ^A -RPO and σ^B -RPO are stable for more than 24 hours. Although heparin disrupt σ^A -RPO and σ^B -RPO quickly, it is not a physiologically relevant competitor. Therefore, we have deleted

the results of heparin challenge assay in the revised manuscript.

8. Figures S4 and S6 - please present 3DFSCs (<https://3dfsc.salk.edu/>) to ensure that the maps are not distorted by the uneven particle orientation distribution.

The 3DFSCs have been added in Figures S4 and S6. The added information is excerpted below:

Supplementary Figure 4. Data validation for σ^A -RPo.

(A) A representative cryo-EM micrograph of σ^A -RPo.

(B) Corrected, masked, unmasked, and phase randomized FSC curves. The dashed line represents the 0.143 FSC cutoff.

(C) Cryo-EM density map colored by local resolution.

(D) Angular distribution of particle projections.

(E) The 3DFSC curve was created on <https://3dfsc.salk.edu/>.

(F) Masked and unmasked map-to-model FSC curves. The dashed line represents the 0.5 FSC cutoff.

Supplementary Figure 6. Data validation for σ^B -RPo.

(A) A representative cryo-EM micrograph of σ^B -RPo.

(B) Corrected, masked, unmasked, and phase randomized FSC curves. The dashed line represents the 0.143 FSC cutoff.

(C) Cryo-EM density map colored by local resolution.

(D) Angular distribution of particle projections.

(E) The 3DFSC curve was created on <https://3dfsc.salk.edu/>.

(F) Masked and unmasked map-to-model FSC curves. The dashed line represents the 0.5 FSC cutoff.

Minor issues

9. It would be informative to include sequence logos (see Shultzaberger et al., 2007; pmid: 1718927) for the sigmaB promoter consensus elements.

Sequence logos have been generated and added in Figure 1.

Reviewer #2:

The manuscript by Yuan et al. presents the first high-resolution structure of the RNA polymerase from *Staphylococcus aureus*, a significant human pathogen. By expressing recombinant polymerase subunits in *E. coli* cells, the authors successfully reconstituted both core and holo SAU RNAP. This achievement represents an important milestone in SAU RNAP research.

The structural data obtained for holoenzyme assembled with different sigma factors (sigma A and B) and promoter DNA, combined with functional assays and mutagenesis, elucidate the mechanism behind the distinct promoter specificities of these transcription factors. The structures also pave the way for structure-assisted studies on SAU RNAP, potentially leading to the development of new antibiotics and thus constitutes a significant contribution to the field. However, the structural findings obtained in this study are not sufficiently analyzed, and the level of insights on the structural basis of promoter recognition is insufficient. The manuscript would further benefit from a detailed analysis of the overall architecture of SAU RNAP and a comparison of the obtained complexes with the available structures of initiation complexes of other well-studied bacterial RNAP. The presentation of the figures and writing should also be improved. A major revision of the manuscript is required to address the issues summarized below:

1. While the authors claim that the obtained recombinant RNAP is active and suitable for structural studies, additional data are needed for justification.

First, the cloning of core SAU RNAP is not described in sufficient detail. Please include the exact pET vector description, and present a map of the obtained plasmid and its complete sequence.

Second, the activity of the core enzyme is not demonstrated. This activity must be measured using a primer extension assay on an RNA-DNA scaffold.

Finally, the activity of the holoenzyme is not demonstrated sufficiently because the transcription assay does not distinguish between run-off and abortive products. It is, therefore, impossible to determine whether the recombinant holoenzyme is capable of productive transcription or if it can only form off-path complexes on promoter DNA.

We would like to thank the reviewer for his/her time in reviewing our manuscript and also for the detailed instructions to improve the quality of the manuscript. The cloning of RNAP has been described in detail on page 13. A map of the obtained plasmid and its complete sequence has been added in Figure S1 and a Supplementary Sequences file, respectively. The activity of the RNAP core enzyme has been verified using a primer extension assay with a 5' 6-FAM labelled RNA. The activity of the holoenzyme has been confirmed by urea-PAGE of the transcript. The gel images have been added in Figure S1. The added information is excerpted below:

Supplementary Figure 1. Purification of *S. aureus* RNAP.

- (A) The map of plasmid pET21a-Sau-rpoABCZEY.
- (B) SDS-PAGE of *S. aureus* σ^A , σ^B , RNAP core and holoenzyme.
- (C) Primer extension assay confirms the activity of *S. aureus* RNAP core enzyme.
- (D) Run-off transcription assay confirms the activity of *S. aureus* RNAP holoenzyme.

2. To better evaluate the quality of the model fitting, the authors should provide map-to-model Fourier Shell Correlation (FSC) plots, highlighting resolution cutoffs at FSC = 0.5. The map-to-model correlation coefficients (CC) should be added to the Table S1.

The map-to-model FSC plots have been added in Figures S4 and S6. The map-to-model correlation coefficients have been added to Table S1 (Table S2 in the revised manuscript). The added information is excerpted below:

Supplementary Figure 4. Data validation for σ^A -RPo.

(A) A representative cryo-EM micrograph of σ^A -RPo.

(B) Corrected, masked, unmasked, and phase randomized FSC curves. The dashed line represents the 0.143 FSC cutoff.

(C) Cryo-EM density map colored by local resolution.

(D) Angular distribution of particle projections.

(E) The 3DFSC curve was created on <https://3dfsc.salk.edu/>.

(F) Masked and unmasked map-to-model FSC curves. The dashed line represents the 0.5 FSC cutoff.

Supplementary Figure 6. Data validation for σ^B -RPo.

(A) A representative cryo-EM micrograph of σ^B -RPo.

(B) Corrected, masked, unmasked, and phase randomized FSC curves. The dashed line represents the 0.143 FSC cutoff.

(C) Cryo-EM density map colored by local resolution.

(D) Angular distribution of particle projections.

(E) The 3DFSC curve was created on <https://3dfsc.salk.edu/>.

(F) Masked and unmasked model-to-map FSC curves. The dashed line represents the 0.5 FSC cutoff.

3. A supporting figure showing the sigma R2/R3 -10 DNA interface from the Rpo-sigma B structure should be provided with the segment of the cryoEM density map. Residues critical for -10 region recognition, such as R74, R100, and R110, should be highlighted.

Density maps showing these interactions have been added in Figure S7. The added information

is excerpted below:

Supplementary Figure 7. Representative electron potential map and superimposed model.

(A) The electron potential map without B-factor sharpening and the superimposed model of σ^A and σ^B . The contour level is 0.01. The carve radius is 3.4 Å.

(B) The electron potential map without B-factor sharpening and the superimposed model of *rrnB* P1 and *yabJ*. The contour level is 0.01. The carve radius is 3.4 Å.

(C) The electron potential map with B-factor sharpening and the superimposed model of σ^B_4 -DNA. The contour level is 0.01. The carve radius is 3.4 Å.

(D) The electron potential map with B-factor sharpening and the superimposed model of σ^A -DNA and σ^B -DNA. The contour level is 0.01. The carve radius is 3.4 Å.

4. The rotamer outliers and clash scores should be improved for PDB 8X6F. The latest versions of Phenix can improve the rotamer outliers.

The rotamer outliers and clash scores have been improved using the latest version of Phenix. The PDB validation report for 8X6F has been updated in the manuscript submission system.

- The manuscript should discuss the overall structure of SAU RNAP, highlighting differences from the previously published structures of bacterial polymerases.

The overall structure and species specific insertion of *S. aureus* RNAP has been discussed on page 4. The added information is excerpted below:

There is only one insertion in *S. aureus* RNAP (β In5, L281-K373, Supplementary Fig. 9). β In5 inserts into and packs against the β lobe, resulting in an interface area of 1617 Å². The large interface area makes β In5 and β lobe look like one whole domain. Since β lobe is the target of transcription factors³⁰⁻³², β In5 may serve as the docking site for transcription factors. In *E. coli* RNAP, β SI1 inserts into the β lobe at a different site. The interface area of β SI1 is much smaller and it is attached to the β lobe loosely.

Supplementary Figure 9. β In5 inserts into and packs against the β lobe.

(A) The overall structure of *S. aureus* σ^A -RPo.

(B) Structural superimposition of *E. coli* σ^{70} -RPo (PDB: 6CA0) and *S. aureus* σ^A -RPo.

- Introduce a figure comparing interactions of the -35 promoter region with Sigma A and B.

A figure comparing interactions of the -35 element with σ^A and σ^B has been added in Figure S10. The added information is excerpted below:

Supplementary Figure 10. Comparison of σ^A_4 -DNA interactions (gray) and σ^B_4 -DNA interactions (colors as in Fig. 2A).

7. Comparison to available structures of bacterial transcription initiation complexes (e.g. Sigma70/ SigA in *E. coli*, and Sigma H in *M. tuberculosis*) should be included in the manuscript. This would help readers understand different modes of sigma factor binding to -10/-35 promoter DNA.

Comparison to available structures of bacterial transcription initiation complexes have been added in the revised manuscript. The added information is excerpted below:

Structural comparison of different σ factors reveals the reason for the short spacers between the -35 element and -10 element of σ^B regulated promoters. Similar to the structure of *E. coli* σ^{70} -RPO¹⁰, the structure of *S. aureus* σ^A -RPO demonstrates that σ^A_4 recognizes the promoter -35 element through its helix-turn-helix (HTH) motif and σ^A_2 recognizes the promoter -10 element through two cognate protein pockets (Fig. 5A). Despite the lack of $\sigma_{1.1}$, $\sigma_{1.2}$, and σ_3 , the structure of *M. tuberculosis* σ^H -RPO shows that σ^H also binds to the promoter in an analogous manner¹². Like σ^{70}/σ^A and σ^H , σ^B_4 recognizes the promoter -35 element through its HTH motif. Unlike σ^{70}/σ^A and σ^H , σ^B_2 and σ^B_3 co-recognize the -10 element. Since σ_2 , σ_3 , and σ_4 are anchored to RNAP surface at the fixed locations, the spacers between the -35 element and -10 element of σ^B regulated promoters are ~3 bp shorter than those of σ^{70}/σ^A and σ^H regulated promoters.

Fig. 5. Structural comparison of *E. coli* σ^{70} -RPO, *M. tuberculosis* σ^H -RPO, *S. aureus* σ^A -RPO, and σ^B -RPO

(A) σ -DNA interactions in *E. coli* σ^{70} -RPO (PDB: 6CA0), *M. tuberculosis* σ^H -RPO (PDB:

5ZX2), *S. aureus* σ^A -RPO, and σ^B -RPO. Yellow, σ ; salmon, nontemplate strand DNA; red, template strand DNA; green, -35 element; blue, -10 element.

(B) Sequence alignment of σ^B orthologs from *S. aureus* (Sau), *Bacillus subtilis* (Bsu), *Bacillus cereus* (Bce), *Clostridium difficile* (Cdi), *Listeria monocytogenes* (Lmo), and *Mycobacterium tuberculosis* (Mtb). The DNA interacting residues of *S. aureus* σ^B are indicated by black triangles.

8. The manuscript will benefit from a more specific title. Perhaps “Structural basis of promoter recognition by *Staphylococcus aureus* RNAP” will do the job.

The title has been changed to “Structural basis of promoter recognition by *Staphylococcus aureus* RNA polymerase”.

9. The significance of delta and epsilon subunits is not sufficiently discussed.

The significance of delta and epsilon subunits has been discussed on page 8. The added information is excerpted below:

Our cryo-EM structures also hints at mechanisms of action for δ subunit during transcription initiation. Our cryo-EM structures demonstrate that the N-terminal domain of δ subunit anchors the C-terminal region at the rim of the main channel, where the C-terminal region can reach into the main channel and exclude the binding of DNA (Supplementary Fig. 8). Accordingly, δ subunit has been shown to interfere with the interaction between DNA and RNAP^{33,34}. Our biochemical experiments indicate that $\sigma^{A_{1.1}}$ also resides in the main channel and exclude the binding of DNA. Therefore, the C-terminal region of δ subunit probably competes with $\sigma^{A_{1.1}}$ and occupies similar regions in the main channel. Consistently, *Bacillus subtilis* δ subunit exhibits negative cooperativity with σ^A and favors its exchange for alternative σ factors that lack $\sigma^{A_{1.1}}$ ⁴⁵⁻⁴⁷.

10. There are no references to Fig 1A and Fig1B in the text of the manuscript.

The references to Fig. 1A and Fig. 1B have been added on page 3.

11. Figure 3A lacks Sigma A region 3, making comparison with Sigma B difficult.

σ^A_3 has been added in Figure 3A. The new figure is excerpted below:

Fig. 3. σ -DNA interactions responsible for -10 element recognition

(A) σ^A_2 -DNA interactions. Yellow, σ ; light blue, nontemplate strand DNA; dark blue, template strand DNA. Left subpanel, ribbon representation; right subpanel, surface representation.

(B) σ^B_2 and σ^B_3 co-recognize the -10 element. Yellow, σ ; light blue, nontemplate strand

DNA; dark blue, template strand DNA. Left subpanel, ribbon representation; right subpanel, surface representation.

(C) EMSA shows that the substitution of DNA interacting residues impairs σ^B -RPO formation.

(D) Ribogreen transcription assay shows that the substitution of DNA interacting residues impairs σ^B dependent transcription. Error bars represent mean \pm SD of $n = 3$ experiments. ****, $p < 0.0001$; one-way ANOVA.

(E) EMSA shows that the mutation of the interacting nucleotides impairs σ^B -RPO formation.

(F) Ribogreen transcription assay shows that the mutation of the interacting nucleotides impairs σ^B dependent transcription. Error bars represent mean \pm SD of $n = 3$ experiments. **, $p < 0.01$; ****, $p < 0.0001$; one-way ANOVA.

(G) Sequence alignment of *S. aureus* σ^A and σ^B . The DNA interacting residues of σ^B are indicated by black triangles.

12. Regions -35 and -10 are highlighted in green and blue in Fig 1B. Keeping these colors consistent with the other figures in the manuscript will be helpful.

The colors of DNA have been modified in the revised manuscript.

13. Statements about the clamp are not illustrated in the manuscript.

The illustration of the clamp has been added in Figure S8. The added information is excerpted below:

Supplementary Figure 8. The clamp adopts a closed conformation, securing the transcription bubble and downstream dsDNA in the main channel.

14. The manuscript will benefit from professional editing to improve grammar and eliminate jargon : Fig3 legend: change “jeopardizes” to “impairs”, “Expressed recombinantly” - to expression of recombinant protein, “Excessive sigma factor” to “an excess of a sigma factor”, etc

The manuscript has been edited to improve grammar and eliminate jargon.

Reviewer #3:

The manuscript by Yuan et al. describes two cryo-EM structures of *Staphylococcus aureus* RNA polymerase in complex with its two sigma factors (sigmaA and sigmaB) and respective promoters. sigmaA is responsible for transcribing house-keeping genes and sigmaB virulence factors. The quality of the structures is difficult to assess as no detailed electron density is shown or described adequately. The manuscript is poorly written and does not provide any mechanistic insights into the action and gene regulation by the two sigma factors. The DNA structures are also poorly built and the models look distorted in figures.

We would like to thank the reviewer for his/her time in reviewing our manuscript and also for the detailed instructions to improve the quality of the manuscript. The manuscript has been revised thoroughly. Electron densities have been added in Figure S7. The DNA structures have been modified to improve geometry.

Main points:

1) The Introduction is superficial. It explains extensively the importance of *S. aureus* and sigmaB on virulence but the results here do not explain how that is achieved so the content is unnecessarily detailed.

The details of *S. aureus* have been deleted as the reviewer suggested.

2) Instead, there is no information on the two sigma factors. What is known biochemically? How are they regulated? Do they need to be activated? How do they convert from closed to open complex?

More information on σ^A and σ^B has been added in the Introduction. The added information is excerpted below:

σ^B is the first alternative σ factor described in bacteria¹⁸. In *S. aureus*, the σ^B gene is part of an operon, formed with *rsbU*, *rsbV*, and *rsbW*. The activity of σ^B is regulated on the post-transcriptional level by the Rsb proteins. σ^B is sequestered in a stable complex with the anti- σ factor RsbW during exponential growth. Binding of the anti-anti- σ factor RsbV leads to the release of σ^B from RsbW and subsequently allows its binding to RNAP. The first gene in the σ^B operon, *rsbU*, was shown to be necessary for the activation of σ^B and is the major activator of σ^B during acidic stress^{19,20}. σ^B directly and indirectly controls approximately 200 genes, including genes with functions in virulence, biofilm formation, persistence, cell internalization, membrane transport, and antimicrobial resistance²¹⁻²³. Although σ^A and σ^B share σ_2 , σ_3 , $\sigma_{3.2}$, and σ_4 , the promoters of σ^B regulated genes show distinct signatures from those of σ^A regulated genes, ensuring the specificity of transcription regulation (Fig. 1A and 1B). First, the consensus sequence of the -35 element (GTTTWW) and -10 element (GGGWAW) are dramatically different from those of σ^A dependent promoters²¹. More importantly, the spacers between the -35 element and -10 element are divergent (~17 bp for σ^A vs ~14 bp for σ^B).

- 3) In addition, functional similarities to other systems including other gram-positive and gram-negative sigma70, ECF sigma and sigma54 should be discussed including consensus sequences and the spacers between the two recognition sites.

The comparison of σ^{70} , σ^H , and σ^{54} has been added in the Introduction. The added information is excerpted below:

Bacterial RNAP forms holoenzyme with σ factors to initiate transcription⁵. Housekeeping σ factors (σ^{70} in *E. coli* and σ^A in other bacteria) govern the transcription of the majority of cellular genes. Housekeeping σ factors are comprised of several conserved domains: $\sigma_{1.1}$, $\sigma_{1.2}$, σ_2 , σ_3 , $\sigma_{3.2}$, and σ_4 . For housekeeping σ factors, the consensus sequences of the promoter -35 element and -10 element are TTGACA and TATAAT, with an optimal spacer of 17 base pairs (bp). Extensive genetic, biochemical and structural studies demonstrate that $\sigma^{70/A}_4$ contacts the flap tip helix (FTH) of the RNAP β subunit and recognizes the promoter -35 element as double-stranded DNA (dsDNA), while $\sigma^{70/A}_2$ contacts the clamp helices of the RNAP β' subunit and recognizes the promoter -10 element as single-stranded DNA (ssDNA)⁶⁻¹⁰. In contrast to the housekeeping σ factors, alternative σ factors direct RNAP to specialized operons in response to environmental and physiological cues. For example, *Mycobacterium tuberculosis* σ^H is a key regulator of the response to oxidative, nitrosative, and heat stresses¹¹. For σ^H regulated promoters, the consensus sequences of the -35 element and -10 element are GGAACA and GTT, with an optimal spacer of 17 bp. Similar to the housekeeping σ factors, the -35 element and -10 element are recognized by σ^H_4 and σ^H_2 as dsDNA and ssDNA, respectively¹². σ^{54} , which is involved in a range of different stress responses, has no sequence similarity to housekeeping σ factors at all¹³⁻¹⁷. In contrast to σ^{70} and σ^H , σ^{54} is unable to unwind promoter DNA spontaneously. Instead, it requires ATP dependent activator proteins bound upstream of the promoter in order to initiate transcription. The consensus sequences of the promoter -24 element and -12 element are TGGCACG and TTGCW (W = A/T), with an optimal spacer of 4 bp. σ^{54} recognizes the promoter -24 element and -12 elements using RpoN and ELH-HTH domains, respectively.

- 4) Structures of RNAP and sigma factors and what are known about them functionally should be explained.

Structures of RNAP and σ factors have been described in the Introduction. The added information is excerpted below:

Bacterial RNA polymerase (RNAP) is the protein machinery responsible for transcription. Most bacterial RNAP is composed of five subunits-- α^I , α^{II} , β , β' , and ω . The overall shape of bacterial RNAP resembles a crab claw, with the active center cleft located in the middle of two pincers¹. During transcription initiation, the clamp, a mobile structural module that makes up much of one pincer, undergoes swing motions that open the active center cleft to allow entry of the promoter DNA²⁻⁴. During transcription elongation, the clamp closes

up and secures the transcription bubble inside the active center cleft.

Bacterial RNAP forms holoenzyme with σ factors to initiate transcription⁵. Housekeeping σ factors (σ^{70} in *E. coli* and σ^A in other bacteria) govern the transcription of the majority of cellular genes. Housekeeping σ factors are comprised of several conserved domains: $\sigma_{1.1}$, $\sigma_{1.2}$, σ_2 , σ_3 , $\sigma_{3.2}$, and σ_4 . For housekeeping σ factors, the consensus sequences of the promoter -35 element and -10 element are TTGACA and TATAAT, with an optimal spacer of 17 base pairs (bp). Extensive genetic, biochemical and structural studies demonstrate that $\sigma^{70/A}_4$ contacts the flap tip helix (FTH) of the RNAP β subunit and recognizes the promoter -35 element as double-stranded DNA (dsDNA), while $\sigma^{70/A}_2$ contacts the clamp helices of the RNAP β' subunit and recognizes the promoter -10 element as single-stranded DNA (ssDNA)⁶⁻¹⁰. In contrast to the housekeeping σ factors, alternative σ factors direct RNAP to specialized operons in response to environmental and physiological cues. For example, *Mycobacterium tuberculosis* σ^H is a key regulator of the response to oxidative, nitrosative, and heat stresses¹¹. For σ^H regulated promoters, the consensus sequences of the -35 element and -10 element are GGAACA and GTT, with an optimal spacer of 17 bp. Similar to the housekeeping σ factors, the -35 element and -10 element are recognized by σ^H_4 and σ^H_2 as dsDNA and ssDNA, respectively¹². σ^{54} , which is involved in a range of different stress responses, has no sequence similarity to housekeeping σ factors at all¹³⁻¹⁷. In contrast to σ^{70} and σ^H , σ^{54} is unable to unwind promoter DNA spontaneously. Instead, it requires ATP dependent activator proteins bound upstream of the promoter in order to initiate transcription. The consensus sequences of the promoter -24 element and -12 element are TGGCACG and TTGCW (W = A/T), with an optimal spacer of 4 bp. σ^{54} recognizes the promoter -24 element and -12 elements using RpoN and ELH-HTH domains, respectively.

5) There are no descriptions or discussions on the RNAP and sigma factors. What are the structural similarities and differences with other known sigma factors and their interactions with DNA?

The comparison of σ^{70} , σ^A , σ^B , and σ^H has been added in the Discussion. The added information is excerpted below:

Structural comparison of different σ factors reveals the reason for the short spacers between the -35 element and -10 element of σ^B regulated promoters. Similar to the structure of *E. coli* σ^{70} -Rpo¹⁰, the structure of *S. aureus* σ^A -Rpo demonstrates that σ^A_4 recognizes the promoter -35 element through its helix-turn-helix (HTH) motif and σ^A_2 recognizes the promoter -10 element through two cognate protein pockets (Fig. 5A). Despite the lack of $\sigma_{1.1}$, $\sigma_{1.2}$, and σ_3 , the structure of *M. tuberculosis* σ^H -Rpo shows that σ^H also binds to the promoter in an analogous manner¹². Like σ^{70}/σ^A and σ^H , σ^B_4 recognizes the promoter -35 element through its HTH motif. Unlike σ^{70}/σ^A and σ^H , σ^B_2 and σ^B_3 co-recognize the -10 element. Since σ_2 , σ_3 , and σ_4 are anchored to RNAP surface at the fixed locations, the spacers between the -35 element and -10 element of σ^B regulated promoters are ~3 bp shorter than those of σ^{70}/σ^A and σ^H regulated promoters.

Fig. 5. Structural comparison of *E. coli* σ^{70} -RPo, *M. tuberculosis* σ^H -RPo, *S. aureus* σ^A -RPo, and σ^B -RPo

(A) σ -DNA interactions in *E. coli* σ^{70} -RPo (PDB: 6CA0), *M. tuberculosis* σ^H -RPo (PDB: 5ZX2), *S. aureus* σ^A -RPo, and σ^B -RPo. Yellow, σ ; salmon, nontemplate strand DNA; red, template strand DNA; green, -35 element; blue, -10 element.

(B) Sequence alignment of σ^B orthologs from *S. aureus* (Sau), *Bacillus subtilis* (Bsu), *Bacillus cereus* (Bce), *Clostridium difficile* (Cdi), *Listeria monocytogenes* (Lmo), and *Mycobacterium tuberculosis* (Mtb). The DNA interacting residues of *S. aureus* σ^B are indicated by black triangles.

6) The main differences between the two structures lie in the fact that in σ^B , region 2 and region 3 of sigma are involved in DNA binding, they argue this causes/explains the shorter spacer. Which is the major DNA binding site? Does Region 4 bind to -35 first, what is the order of binding?

According to the study of *E. coli* σ^{70} , the -35 element dsDNA binds to σ^{70}_4 first. Then the -10 element dsDNA unwinds and binds to σ^{70}_2 . *S. aureus* σ^A probably works in the same way as *E. coli* σ^{70} . We don't know the exact order of binding for *S. aureus* σ^B . A reasonable speculation is that -35 element and -10 element dsDNA bind to σ^B_4 , σ^B_3 , and σ^B_2 simultaneously. Then σ^B_5 of the -10 element flips out and inserts into a pocket of σ^B_2 . The discussion has been added on page 7. The added information is excerpted below:

The conversion from RNAP-promoter closed complex (RPc) to RPo has been studied

extensively using *E. coli* σ^{70} ^{30,31,39}. In σ^{70} -RPO, sequence specific recognition of the promoter -35 element by σ_4 positions the critical and conserved A₂ of -10 element in line with σ_2 residues that later capture the flipped base to nucleate transcription bubble formation. In σ^{70} -RPO, two conserved pockets in σ^{70} capture the flipped bases of the -10 element (A₂ and T₆) and stabilize the transcription bubble. *S. aureus* σ^A probably works in the same way as *E. coli* σ^{70} . As for σ^B , sequence specific recognition of the promoter -35 element and -10 element by σ_4 , σ_3 , and σ_2 positions the conserved A₅ of -10 element in line with σ_2 residues that later capture the flipped base to nucleate transcription bubble formation. Since there is no structural equivalent of the T₆ pocket of σ^{70}/σ^A , only one base of the -10 element (A₅) is flipped and specifically captured in a protein pocket.

7) What is the role of region 1 in sigmaA ? Are they similar to sigma70 ? What are the similarities and differences

Previous studies showed that *E. coli* $\sigma^{70}_{1.1}$ modulates the DNA binding activity of σ^{70} . In the absence of RNAP, $\sigma^{70}_{1.1}$ inhibits the DNA binding function of free σ^{70} . In the presence of RNAP, $\sigma^{70}_{1.1}$ binds in the main channel of RNAP and prevents the nonspecific binding of DNA. There is no density for *S. aureus* $\sigma^A_{1.1}$ in the structure of σ^A -RPO, but the structure of *S. aureus* $\sigma^A_{1.1}$ predicted by AlphaFold is very similar to the structure of *E. coli* $\sigma^{70}_{1.1}$ and *B. subtilis* $\sigma^A_{1.1}$, suggesting their similar roles. To delineate the function of *S. aureus* $\sigma^A_{1.1}$, we constructed and purified $\sigma^A_{1.1}$ truncated σ^A . Fluorescence polarization experiments demonstrate that $\sigma^A_{1.1}$ truncated σ^A binds promoter DNA better than the full-length σ^A . Moreover, truncation of $\sigma^A_{1.1}$ increases σ^A dependent RPO formation, confirming that the roles of *E. coli* $\sigma^{70}_{1.1}$ and *S. aureus* $\sigma^A_{1.1}$ are similar. The predicted structure of *S. aureus* $\sigma^A_{1.1}$ and the results of fluorescence polarization assay and EMSA have been added in Figure 4 and discussed on page 6. The added information is excerpted below:

Fig. 4. *S. aureus* $\sigma^{A}_{1.1}$ suppresses the DNA binding activity of free σ^A and σ^A -RNAP holoenzyme

(A) Structural comparison of *S. aureus* $\sigma^{A}_{1.1}$, *E. coli* $\sigma^{70}_{1.1}$ (PDB: 4LK1), and *B. subtilis* $\sigma^{A}_{1.1}$ (PDB: 5MWW). The structure of *S. aureus* $\sigma^{A}_{1.1}$ is predicted by AlphaFold.

(B) Fluorescence polarization assay shows that $\sigma^{A}_{1.1}$ truncated σ^A binds promoter DNA better than the full-length σ^A . Error bars represent mean \pm SD of n = 3 experiments. **, $p < 0.01$; ***, $p < 0.001$; two-tailed Student's *t*-test.

(C) EMSA shows that truncation of $\sigma^{A}_{1.1}$ increases σ^A -dependent RPo formation.

8) Can sigmaA and sigmaB bind to DNA in the absence of RNAP ?

A fluorescence polarization assay has been performed to test whether σ^A and σ^B bind to DNA in the absence of RNAP. σ^A and σ^B bind to DNA if high concentrations of protein are used. Truncation of $\sigma^{A}_{1.1}$ further improves the DNA binding activity of σ^A . The results of fluorescence polarization assay have been added in Figure 4 and discussed on page 6.

9) Do they see density for transcription bubble ? what causes the conversion from closed to open complex in their sample preparation ?

There is density for the nontemplate strand ssDNA of the transcription bubble in σ^A -RPo. There is density for the first nucleotide of the nontemplate strand ssDNA of the transcription bubble in σ^B -RPo. Density for other nucleotides of the transcription bubble is weak. DNA is unwound in our sample preparation because 1) the DNA scaffolds contain the consensus -35 and -10 elements, 2) a high concentration of RNAP was used for cryo-EM sample preparation, and 3) the cryo-EM samples were incubated at 37°C for 10 min.

Other points:

1) Fig. S1, lanes containing sB and RNAP core samples have multiple other bands. What are they ?

Because the yields of *S. aureus* σ^B and RNAP are low, their purity is worse than that of *E. coli*. The contamination was removed by 2D and 3D classification during cryo-EM data processing.

2) Detailed electron density should be shown for DNA and regions of protein-DNA interactions to provide confidence in the quality of the reconstructions.

Density maps showing protein-DNA interactions have been added in Figure S7. The added information is excerpted below:

Supplementary Figure 7. Representative electron potential map and superimposed model.

(A) The electron potential map without B-factor sharpening and the superimposed model of σ^A and σ^B . The contour level is 0.01. The carve radius is 3.4 Å.

(B) The electron potential map without B-factor sharpening and the superimposed model of *rrnB* P1 and *yabJ*. The contour level is 0.01. The carve radius is 3.4 Å.

(C) The electron potential map with B-factor sharpening and the superimposed model of σ^B_4 -DNA. The contour level is 0.01. The carve radius is 3.4 Å.

(D) The electron potential map with B-factor sharpening and the superimposed model of σ^A -DNA and σ^B -DNA. The contour level is 0.01. The carve radius is 3.4 Å.

3) FSC for masked, unmasked and phase randomised should also be shown.

FSC for masked, unmasked and phase randomized have been added in Figures S4 and S6. The added information is excerpted below:

Supplementary Figure 4. Data validation for σ^A -RPo.

(A) A representative cryo-EM micrograph of σ^A -RPo.

(B) Corrected, masked, unmasked, and phase randomized FSC curves. The dashed line represents the 0.143 FSC cutoff.

(C) Cryo-EM density map colored by local resolution.

(D) Angular distribution of particle projections.

(E) The 3DFSC curve was created on <https://3dfsc.salk.edu/>.

(F) Masked and unmasked map-to-model FSC curves. The dashed line represents the 0.5 FSC cutoff.

Supplementary Figure 6. Data validation for σ^B -RPo.

(A) A representative cryo-EM micrograph of σ^B -RPo.

(B) Corrected, masked, unmasked, and phase randomized FSC curves. The dashed line represents the 0.143 FSC cutoff.

(C) Cryo-EM density map colored by local resolution.

(D) Angular distribution of particle projections.

(E) The 3DFSC curve was created on <https://3dfsc.salk.edu/>.

(F) Masked and unmasked map-to-model FSC curves. The dashed line represents the 0.5 FSC cutoff.

4) Figures are overall poor. Figures should show different regions of sigma regions, accomplished by density in regions that show detailed interactions.

Some figures have been modified for clarity. Density maps have been added in Figure S7.

5) Fig. 2, B and D, for WT, why are the two EMSA result different? Same is true for Fig. 3 C and E.

The experiments have been repeated. The figures have been replaced with more consistent results. The new figures are excerpted below:

Fig. 2. σ -DNA interactions responsible for -35 element recognition

(A) σ^B_4 -DNA interactions are depicted in stereo view. Yellow, σ^B_4 ; dark green, nontemplate strand DNA; light green, template strand DNA. The potential hydrogen bonds are shown as dashed lines.

(B) EMSA shows that the substitution of DNA interacting residues impairs σ^B -RPo formation.

(C) Ribogreen transcription assay shows that the substitution of DNA interacting residues impairs σ^B dependent transcription. Error bars represent mean \pm SD of n = 3 experiments. *, $p < 0.05$; ***, $p < 0.001$; ****, $p < 0.0001$; one-way ANOVA.

(D) EMSA shows that the mutation of the interacting nucleotides impairs σ^B -RPo formation.

(E) Ribogreen transcription assay shows that the mutation of the interacting nucleotides impairs σ^B dependent transcription. Error bars represent mean \pm SD of n = 3 experiments. ****, $p < 0.0001$; one-way ANOVA.

(F) Sequence alignment of *S. aureus* σ^A and σ^B . The DNA interacting residues of σ^B are indicated by black triangles.

Fig. 3. σ -DNA interactions responsible for -10 element recognition

(A) σ^A_2 -DNA interactions. Yellow, σ ; light blue, nontemplate strand DNA; dark blue, template strand DNA. Left subpanel, ribbon representation; right subpanel, surface representation.

(B) σ^B_2 and σ^B_3 co-recognize the -10 element. Yellow, σ ; light blue, nontemplate strand

DNA; dark blue, template strand DNA. Left subpanel, ribbon representation; right subpanel, surface representation.

(C) EMSA shows that the substitution of DNA interacting residues impairs σ^B -RPO formation.

(D) Ribogreen transcription assay shows that the substitution of DNA interacting residues impairs σ^B dependent transcription. Error bars represent mean \pm SD of n = 3 experiments. ****, $p < 0.0001$; one-way ANOVA.

(E) EMSA shows that the mutation of the interacting nucleotides impairs σ^B -RPO formation.

(F) Ribogreen transcription assay shows that the mutation of the interacting nucleotides impairs σ^B dependent transcription. Error bars represent mean \pm SD of n = 3 experiments. **, $p < 0.01$; ****, $p < 0.0001$; one-way ANOVA.

(G) Sequence alignment of *S. aureus* σ^A and σ^B . The DNA interacting residues of σ^B are indicated by black triangles.

- 6) Fig. 4B, sigmaB-RPO is less stable to start with. This can't be explained by the lack of region 1 as in sigmaB has additional interactions with region 3.

The relevant statement has been deleted in the revised manuscript.

REVIEWERS' COMMENTS

Reviewer #1 (Remarks to the Author):

The authors have improved the manuscript by responding to the reviewer comments - it is ready for publication

Reviewer #2 (Remarks to the Author):

The authors have clearly addressed all the points raised during review. The revised manuscript is significantly improved and, in my opinion, meets the high standards for publication in Nature Communications.

Reviewer #3 (Remarks to the Author):

The revised manuscript has significantly improved and addressed almost all my concerns. A few minor changes/clarification are needed:

Summary - please reword "Therefore, it is a mystery how σ_B recognizes and initiates transcription from target promoters." To "therefore how sigmaB recognised and intiaites transcription from target promoters can not be inferred from that of the well studied sigma".

Page 3: line 2: "regulated at the post transcription level" – this is unclear as it could be regulated at the mRNA level. Do you mean at protein level ? If so, change to "is regulated by Rsb proteins". How does RsbU activate sigmaB ?

Page 3: last sentence before Results: "define the interactions..... and promoter unwinding in transcription initiation". There are no sufficient data presented to explain promoter unwinding. Please reword "define the interactions... and explain the promoter specificity and the stabilisation of transcription bubble"

Page 8: Last sentence "structure-based discovery of..." How would this work provide that ? Are there known mutations that are resistant to rifampin ? If so, it will be informative to dock rifampin into the structure and map the mutations found in resistant strains on the structure, thus to providing a structural basis for these resistant mutations.

The DNA nucleotide numbering is very confusing. Please use same numbering as in Fig. 1C (-35, -34, -9 etc instead of 1, 2, 3...).

The conversion to RPo seems inefficient while all the useful particles in the datasets are RPo. Please explain.

Responses to Reviewers' Comments

Reviewer #1:

The authors have improved the manuscript by responding to the reviewer comments - it is ready for publication

We would like to thank the reviewer for his/her time in reviewing our manuscript and also for the detailed instructions to improve the quality of the manuscript.

Reviewer #2

The authors have clearly addressed all the points raised during review. The revised manuscript is significantly improved and, in my opinion, meets the high standards for publication in Nature Communications.

We would like to thank the reviewer for his/her time in reviewing our manuscript and also for the detailed instructions to improve the quality of the manuscript.

Reviewer #3

The revised manuscript has significantly improved and addressed almost all my concerns. A few minor changes/clarification are needed:

We would like to thank the reviewer for his/her time in reviewing our manuscript and also for the detailed instructions to improve the quality of the manuscript.

Summary - please reword “Therefore, it is a mystery how σ^B recognizes and initiates transcription from target promoters.” To “therefore how sigmaB recognised and intiaites transcription from target promoters can not be inferred from that of the well studied sigma”.

The Summary has been reworded as the reviewer suggested.

Page 3: line 2: “regulated at the post transcription level” – this is unclear as it could be regulated at the mRNA level. Do you mean at protein level? If so, change to “is regulated by Rsb proteins”.

The sentence has been modified as the reviewer suggested.

How does RsbU activate sigmaB ?

The phosphatase RsbU may activate σ^B by dephosphorylating the anti-anti- σ factor RsbV.

Page 3: last sentence before Results: “define the interactions..... and promoter unwinding in transcription initiation”. There are no sufficient data presented to explain promoter unwinding. Please reword “define the interactions... and explain the promoter specificity and the stabilisation of transcription bubble”

The sentence has been reworded as the reviewer suggested.

Page 8: Last sentence “structure-based discovery of...” How would this work provide that ? Are there known mutations that are resistant to rifampin ? If so, it will be informative to dock rifampin into the structure and map the mutations found in resistant strains on the structure, thus to providing a structural basis for these resistant mutations.

There indeed are known mutations that are resistant to rifampin. After we dock rifampin into the structure, we find that all resistant mutations are positioned within 10 Å from rifampin. Some of them even directly contact rifampin. For example, substitution of β residue H481 would be expected to disrupt two hydrogen bonds between RNAP and rifampin. The docked structure has been added in Supplementary Figure 12.

The DNA nucleotide numbering is very confusing. Please use same numbering as in Fig. 1C (-35, -34, -9 etc instead of 1, 2, 3...).

The DNA nucleotide numbering has been changed as the reviewer suggested.

The conversion to RPo seems inefficient while all the useful particles in the datasets are RPo. Please explain.

There are several possible reasons. First, the conversion to RPo may be more efficient at the RNAP concentration (1 μ M) for cryo-EM sample preparation. Second, free RNAP may be more prone to fall apart due to surface tension during cryo-EM sample preparation. Third, free RNAP may be too flexible to give a reasonable reconstruction during 3D classification.